# VideoExplorer: Boosting Long Video Understanding with Dynamic Temporal Grounding

## Abstract

Long-video understanding (LVU) is a challenging problem in computer vision. Existing methods either downsample frames for single-pass reasoning, sacrificing fine-grained details, or depend on textual reasoning over task-agnostic representations, hindering task-specific perception and exploration. In this paper, we propose **VideoExplorer**, a framework grounded in the principle of "thinking with video", which naturally intertwines planning, temporal grounding, and scalable perception into a coherent reasoning process. Rather than reasoning over a static context, VideoExplorer iteratively formulates sub-questions, locates relevant moments, and performs task-oriented, temporally scalable video understanding until reaching the final answer, enabling faithful, efficient, and interpretable reasoning. To address the lack of LVU training resources, we construct a long-video reasoning dataset using difficulty-adaptive sampling to ensure high-quality trajectories on complex tasks. Building on this dataset, we design a two-stage training pipeline: supervised trajectory initialization followed by trajectory-level preference optimization, encouraging adaptive temporal grounding and iterative information integration guided by downstream rewards. Extensive evaluations on popular long-video understanding and reasoning benchmarks demonstrate VideoExplorer's significant advantage over existing baselines, highlighting its robustness, adaptability, and efficiency. Our code is available in this repository [1].

## 1 Introduction

Long video understanding is crucial for real-world applications such as autonomous driving, embodied agents, and large-scale surveillance. However, it poses significant challenges for current visual-language models (VLMs) Wang et al. (2024a); Bai et al. (2025); Zhu and etc (2025); Lin et al. (2023) due to the extensive input video frames. Modern VLMs Bai et al. (2025); Lin et al. (2023) address this by heavily downsampling frames and applying single-pass reasoning, which inevitably sacrifices fine-grained details and compromises reasoning accuracy. To mitigate the limitations from brute-force downsampling, recent agentic frameworks Wang et al. (2024b; 2025); Tian et al. (2025); Luo et al. (2024); Zhi et al. (2025) augment the reasoning process with retrieval over preprocessed textual assets (e.g., dense captions, object trajectories, audio transitions, etc.), allowing the model to retrieve and incorporate task-specific textual evidence during iterative inference. However, as this preprocessing is task-agnostic, the reasoning process remains constrained to static and lossy textual representations, which are often computationally expensive, thereby precluding direct, task-adaptive perception of the raw video stream.

To address to the above problems, we introduce the principle of "**thinking with video**". Analogue to "thinking with images", where visual models can zoom in, zoom out, or focus on specific regions adaptively, "thinking with video" treats reasoning as a dynamic process of temporally grounded exploration. Instead of reasoning over a static context, "thinking with video" allows the model to interact directly with the raw video. At each step, it decides *what to look for*, *where to watch*, and *at what temporal scale*, flexibly combining on-demand visual perception, including both fine-grained inspection and coarse-grained holistic browsing. Through this iterative process, the model builds a coherent reasoning chain that integrates localized evidence into a task-driven, temporally grounded understanding, enabling faithful, scalable, and interpretable long-video reasoning.

---

[1] https://anonymous.4open.science/r/Video-DeepResearch-CCEF

Building on this principle, we propose **VideoExplorer**, a framework that unifies reasoning with dynamic temporal grounding and task-aware perceptual exploration. Given a task, VideoExplorer incrementally decomposes it into sequential sub-tasks, iteratively identifies relevant temporal segments, and attends to video content at the granularity required for each step. A planner generates intermediate information needs and coordinates reasoning, while a temporal grounder retrieves and verifies candidate segments from long videos, ensuring accurate and concise evidence. The system then performs temporal-scalable video understanding, dynamically adjusting frame sampling according to task demands, before producing the final answer. By integrating these components into a cohesive reasoning loop, VideoExplorer enables efficient, interpretable, and fine-grained long-video understanding that scales to complex tasks without unnecessary computation.

To train VideoExplorer for effective long-video reasoning, we first construct a reasoning-centric dataset with difficulty-adaptive sampling, generating multi-step reasoning trajectories and emphasizing challenging cases to ensure rich supervision beyond simple QA pairs. Based on this dataset, we adopt a two-stage optimization strategy. First, structured imitation via supervised fine-tuning (SFT) teaches the planner and temporal grounder to reproduce expert reasoning traces in addition to final answers. Then, trajectory-level direct preference optimization (TDPO) evaluates complete reasoning trajectories including intermediate steps and rewards faithful reasoning while penalizing flawed paths. This combination enables VideoExplorer to learn interpretable, task-aligned multi-step reasoning for robust performance on complex long-video tasks. We conduct extensive experiments on widely-used long video datasets, such as MLVU and LVBench for general understanding tasks, as well as VideoMMMU and MH-NIAH for complex reasoning tasks. The experimental results demonstrate the effectiveness of our method. In summary, our contributions are threefold:

(1). We introduce the concept of "thinking with video" and propose **VideoExplorer**, which integrates planning, temporal grounding, and scalable video perception into a unified reasoning paradigm.

(2). We construct a **reasoning-centric dataset** and design a **two-stage optimization pipeline** combining structured imitation and trajectory-level preference alignment.

(3). We conduct extensive experiments across diverse long-video benchmarks, demonstrating the scalability, accuracy, and interpretability of our approach compared to existing methods.

## 2 RELATED WORK

### 2.1 MULTI-MODAL LARGE LANGUAGE MODELS

Advancements in large language models (LLMs) (Brown et al., 2020) have significantly advanced visual understanding by incorporating visual features, leading to the emergence of Multi-modal Large Language Models (MLLMs) (Wang et al., 2024a; Chen et al., 2024a; Liu et al., 2023; OpenAI, 2024; Bai et al., 2025; Zhu and etc, 2025; Yuanhan Zhang et al., 2024; Lin et al., 2023; Zhang et al., 2024; Chen et al., 2024b). These models are designed for general-purpose vision tasks and have also demonstrated strong performance in video understanding. However, most existing MLLMs primarily focus on short-context comprehension, typically handling only a few images or short video clips at a time, which limits their effectiveness on long video understanding tasks (Fu et al., 2024; Zhou et al., 2024). To address this limitation, recent efforts have explored more efficient architectures to extend the input context (Xue et al., 2024; Shu et al., 2024; Liu et al., 2025), enabling the processing of longer videos. Nonetheless, the length of input context that these models can handle remains restricted, and understanding videos of hour-long durations remains a significant challenge. In this work, we propose VideoExplorer, which adaptively retrieves task-relevant visual context in the reasoning chain, avoiding accuracy loss from brute-force frame sampling.

### 2.2 LARGE REASONING MODELS

Recent advancements have significantly enhanced the reasoning capabilities of large language models. OpenAI's o1 OpenAI (2025) demonstrated test-time scaling effects, enabling free-form reasoning prior to generating answers and achieving substantial performance gains. DeepSeek-R1 Team (2025) further incentivizes reasoning through a reinforcement learning (RL) algorithm called group robust preference optimization, which automates the generation of high-quality reasoning trajectories via trial-and-error search without extensive human oversight. Building on these insights, several

studies Feng et al. (2025); Chen et al. (2025); Zhang et al. (2025) have explored reasoning in video understanding. These works train models to generate reasoning processes before producing final answers and optimize them with RL algorithms. However, they operate solely on static down-sampled video frames, leading to significant information loss in long-video understanding. Motivated by these findings, we introduce the principle of "thinking with video", enabling models to not only reason over contextual information but also interact with video streams through on-demand temporal grounding.

## 2.3 Agentic Frameworks for Long Video Understanding

Leveraging the powerful capabilities of large language models OpenAI (2023); Brown et al. (2020), recent studies have explored agentic approaches to tackle the long video understanding challenges Wang et al. (2024b); Zhi et al. (2025); Kugo et al. (2025); Cao et al. (2025). However, these approaches transit videos into task-agnostic textual representations (e.g. object locations, dense captions, audio transitions, etc.), and then leverage powerful LLMs to reasoning over the textual representations. This paradigm will inevitably lose the rich visual information in original long videos, leading to sub-optimal performance. Additionally, many of these methods are designed for limited scenarios such as egocentric videos Wang et al. (2024b); Tian et al. (2025), deducing their applicability to more general real-world settings. Building on this perspective, we present VideoExplorer, a new framework grounded in "thinking with video" that supports efficient, scalable, and adaptive long-video understanding.

## 3 Methodology

### 3.1 Preliminary of Long Video Understanding

Long-video reasoning requires a model to comprehend an extended visual sequence $v$ in order to answer a query $q$. This process is formally defined as:

$$\mathcal{Y} = \Phi(v, q), \tag{1}$$

where $\mathcal{Y}$ denotes the generated answer. Ideally, the model could ingest the entire video at once and answer questions based on the complete content. However, in practice, due to the input length constraints of models, it is often difficult to process the entire video in one go. To address this issue, modern visual language models often perform brute-force downsampling of $v$ into a shorter sequence $v' \ll v$, and then reason over the sampled $v'$: $\mathcal{Y} = \Phi(v', q)$. While this approach is straightforward and sufficient for short videos, it becomes problematic for hour-long videos, which typically contain tens of thousands of raw frames. Severe temporal compression inevitably sacrifices fine-grained details, leading to reasoning based on insufficient context and degraded accuracy.

**Agentic frameworks with retrieval augmentation** present a promising alternative. Rather than downsampling the raw video video and reasoning in a single pass, these frameworks iteratively expand the available context through retrieval augmentation. Specifically, these approaches typically preprocess the video into a fixed set of textual representations (e.g. dense captions, object locations, etc.) and construct a retrieval index based on them, denoted as $\mathcal{D}_v$. During inference, a textual reasoning model $\Theta$ analyzes the task, generates sub-queries, retrieves relevant information from the index, and continues reasoning until an answer is obtained. This process can be formulated as:

$$\mathcal{D}_v = \Gamma_{\text{preprocess}}(v), \tag{2}$$

$$\mathcal{Y} = \Theta\left(q \mid \{\mathcal{K}_t\}_{t=1}^T\right), \quad \mathcal{K}_t = \Gamma_{\text{retrieve}}(q_t^{\text{sub}} \mid \mathcal{D}_v), \tag{3}$$

where $q_t^{\text{sub}}$ represents the sub-queries generated by the reasoning model. This iterative retrieval-augmented approach offers a more scalable solution for long video reasoning compared to single-pass methods, as it alleviates the problem of insufficient context.

This iterative retrieval alleviates context limits and scales better than single-pass methods. However, task-agnostic preprocessing compresses rich video signals into generic text, causing information loss. Moreover, such preprocessing is computationally heavy, limiting scalability in practice.

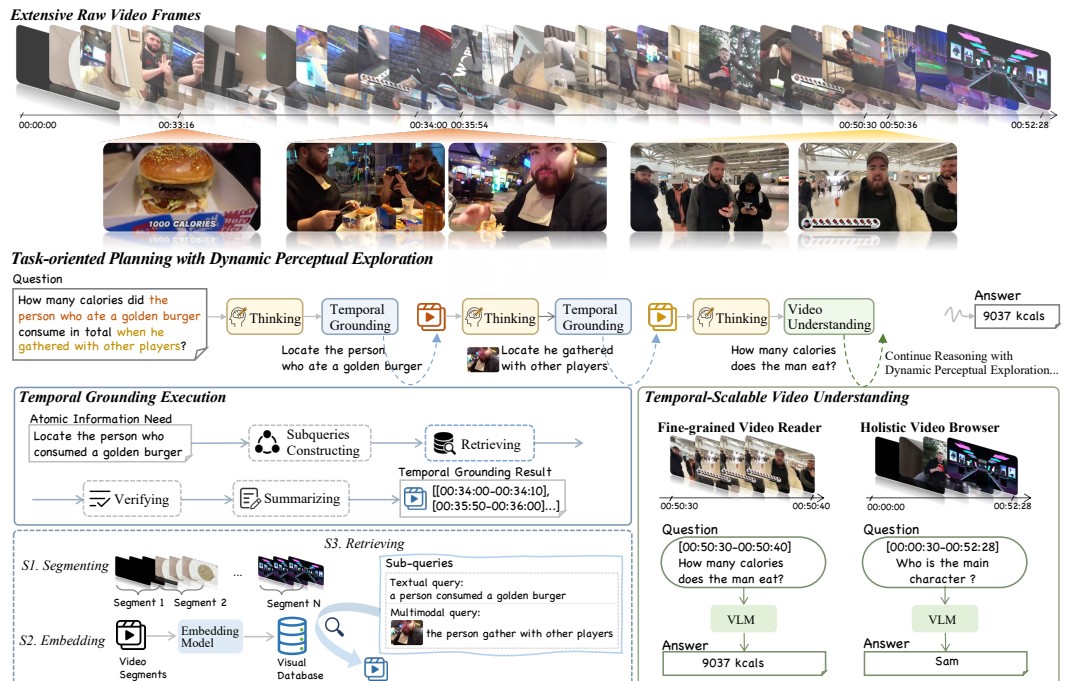

Figure 1: Illustration of VideoExplorer framework. Instead of reasoning over brute-force downsampled frames or static preprocessed database, VideoExplorer integrates planning, temporal grounding, and scalable video perception into a unified reasoning paradigm, where the planner decomposes complex tasks into sub-questions, the temporal grounder adaptively localizes relevant temporal spans, and the perception module dynamically adjusts granularity to meet task demands, enabling faithful, efficient and interpretable long-video understanding.

## 3.2 THINKING WITH VIDEO

Inspired by the idea of "thinking with images", where models can adaptively zoom in, zoom out, or focus on specific regions for a task, we introduce "thinking with video". This paradigm treats reasoning as a dynamic process over on-demand temporal context, where the model raises temporally scalable perceptual queries tailored to sub-tasks, ultimately arriving at the final answer. Rather than relying on brute-force downsampling or task-agnostic preprocessing, "thinking with video" retains direct access to the raw video and adaptively gathers task-relevant information through task-aware temporal grounding and scalable video understanding. Formally, the process is defined as:

$$\mathcal{Y} = \Theta\left(q \mid \{\mathcal{P}_t\}_{t=1}^T\right), \quad \mathcal{P}_t = \Phi_{\text{perc}}(\tau_t), \quad \tau_t = \Gamma_{\text{ground}}(q_t^{\text{sub}} \mid v), \tag{4}$$

where sub-questions $q_t^{\text{sub}}$ are generated by a planner, $\Gamma_{\text{ground}}$ maps them to temporal spans $\tau_t$, and $\Phi_{\text{perc}}$ performs on-demand perception within these spans. This iterative mechanism enables step-by-step reasoning by selectively accessing and processing task-relevant segments of the raw video, rather than compressing or statically pre-processing it.

## 3.3 OUR PROPOSED METHOD: VIDEOEXPLORER

Building on the principle of "thinking with video", we introduce the **VideoExplorer** framework, which achieves scalable long-video reasoning by unifying the reasoning process with dynamic temporal grounding. As illustrated in Figure 1, the generation process unfolds as a sequential composition of cognitive blocks: a thinking block captures intermediate reasoning, a temporal grounding block localizes relevant temporal spans, a video understanding block performs temporal scalable video understanding and return results, and finally an answer block delivers the final response.

**Task-oriented Planning with Dynamic Perceptual Exploration.** The planner, analogous to the human central nervous system, is responsible for analyzing the problem, raising iterative information

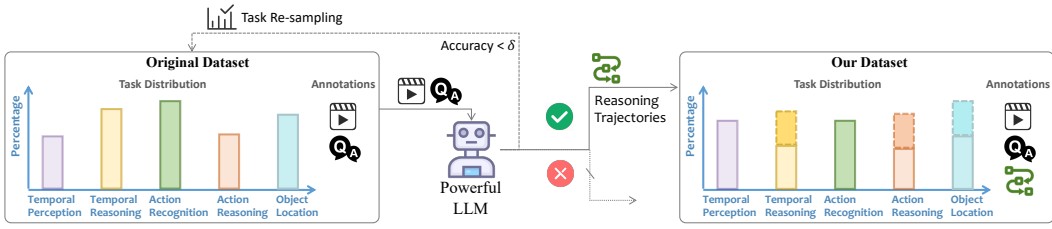

Figure 2: Difficulty-adaptive dataset generation. Tasks are uniformly sampled, reasoning trajectories generated by VideoExplorer, and hard cases re-sampled by first-round accuracy. Only correct-answer trajectories are retained, yielding faithful reasoning trajectories and challenging training data.

needs, reasoning over the current context, and generating the final answer. Instead of passively receiving a fixed set of video frames, the planner can decompose complex tasks, sequentially raise sub-questions, and attach the necessary video segments at controllable temporal granularity, serving as a higher-level controller that guides the overall reasoning trajectory.

**Decoupled Temporal Grounding Execution.** While the planner specifies required information, precise retrieval from long videos remains difficult due to query ambiguity and redundant searches. To reduce reasoning complexity, we introduce a temporal grounder which decouples the retrieval process from the main reasoning chain. In the offline stage, videos are segmented into clips and stored as multimodal embeddings. At inference, the grounding agent matches queries against candidate segments, supporting both text-only and multimodal queries (as shown in the Figure 1). It transforms abstract sub-queries into concrete actions such as construction, retrieval, verification, and summarization, and returns validated segments to the planner. For verification, the temporal grounder leverages VLMs as accurate verifiers to assess whether retrieved segments are relevant. Irrelevant results are discarded, while validated segments are summarized and passed back to the planner. This decoupling enables lightweight reasoning, ensures accurate evidence localization, and mitigates hallucinations from irrelevant segments, allowing the planner to focus on high-level tasks.

**Temporal-Scalable Video Understanding.** Different tasks require different levels of visual granularity. The planner adapts by specifying temporal intervals for perception. For fine-grained queries, it focuses on task-oriented segments located by temporal grounder or specified by the task (e.g., "What happens between 01:20-01:40?"), during which the VLM densely samples frames to obtain accurate answers; for global tasks (e.g., "Who is the main character?"), it covers broader intervals with coarse sampling to support holistic reasoning. This dynamic adjustment balances scalability and fidelity, enabling efficient context management within limited window length while preserving access to high-resolution details when needed.

In summary, VideoExplorer unifies reasoning with dynamic temporal grounding to enable scalable long-video understanding. It features a planner that decomposes complex tasks into iterative sub-questions, a decoupled temporal grounder that localizes and verifies relevant video spans, and a temporal-scalable video understanding module that adapts perception granularity to task demands. Together, these components operate in a coordinated reasoning loop, allowing VideoExplorer to progressively decompose tasks, ground relevant video segments, and adapt perceptual focus, resulting in faithful, efficient, and interpretable long-video reasoning.

### 3.4 Multi-stage Optimization

To enable effective long-video reasoning, we construct a reasoning-centric dataset with difficulty-adaptive sampling, ensuring soft supervising beyond simple outcome supervision and coverage of complex cases. Building on this foundation, we propose a two-stage training framework: (1) structured imitation via supervised fine-tuning (SFT) to instill expert reasoning patterns, and (2) trajectory-level direct preference optimization (TDPO) to refine multi-step reasoning by rewarding faithful trajectories while penalizing flawed ones.

**Constructing a Reasoning-Centric Dataset for Long Video Understanding.** A major challenge in long-video reasoning is the lack of high-quality, reasoning-oriented training data. To address this, we first select a diverse set of long videos from existing datasets, considering video length, source,

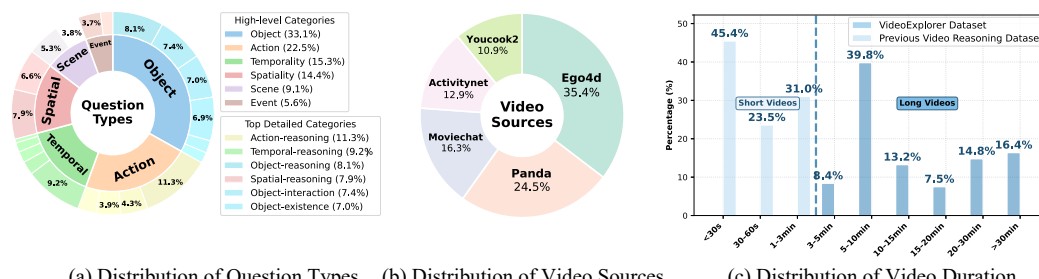

(a).Distribution of Question Types     (b).Distribution of Video Sources     (c).Distribution of Video Duration

Figure 3: Data statistics of VideoExplorer dataset.

and task type. However, these datasets only provide QA labels, which are insufficient for effective training. To enrich them, we leverage a powerful LLM Team (2025) with VideoExplorer to generate additional reasoning data. Since many of the generated tasks are relatively simple, we introduce difficulty-adaptive sampling to emphasize complex reasoning. As illustrated in Figure 2, tasks are uniformly sampled, reasoning trajectories are generated by the LLM with VideoExplorer, and hard cases are re-sampled based on first-round accuracy to construct a more challenging dataset. Only trajectories with correct answers are retained to ensure validity. In total, we curate 11.1k planner trajectories and 10.8k grounding trajectories, with detailed dataset statistics provided in Figure 3.

**Stage I: Structured Imitation via SFT.** We initialize both the planner and the temporal grounding agent through supervised fine-Tuning on expert trajectories. Unlike conventional direct QA supervision, this stage emphasizes structural imitation, where the model learns to reproduce expert reasoning traces in addition to final answers. The objective is the standard negative log-likelihood:

$$\mathcal{L}_{\text{SFT}}(\theta) = -\mathbb{E}(v, q, \mathcal{Y}) \sim \mathcal{D}_{\text{SFT}} \left[\log \pi_\theta(\mathcal{Y} \mid v, q)\right], \tag{5}$$

where $\mathcal{D}_{\text{SFT}}$ includes both answer labels and their associated expert reasoning trajectories. This step enables fast initialization by grounding the model's behavior in high-quality, interpretable reasoning.

**Stage II: Trajectory-Level Preference Optimization.** While SFT provides a fast warmup, it is inherently limited by teacher-forced demonstrations. To move beyond imitation, we introduce trajectory-based direct preference optimization (TDPO), which adapts the DPO framework to the multi-turn and multi-step nature of video reasoning. Instead of evaluating individual responses, TDPO considers entire reasoning trajectories, capturing both intermediate steps and final outcomes. By comparing chosen and rejected trajectories, it encourages the policy to align faithful reasoning with correct task results. Formally, for trajectories $\mathbf{y}_w$ and $\mathbf{y}_l$, the TDPO objective is:

$$\mathcal{L}_{\text{TDPO}}(\pi_\theta; \pi_{\text{ref}}) = -\mathbb{E}_{(\mathbf{y}_w, \mathbf{y}l) \sim \mathcal{D}} \left[\log \sigma \left(\beta \left(\log \frac{\pi_\theta(\mathbf{y}_w)}{\pi_{\text{ref}}(\mathbf{y}_w)} - \log \frac{\pi_\theta(\mathbf{y}_l)}{\pi_{\text{ref}}(\mathbf{y}_l)}\right)\right)\right] \tag{6}$$

Here, $\pi_\theta$ is the learnable policy, $\pi_{\text{ref}}$ is reference policy, $\beta$ controls the preference sharpness, and the expectation is over trajectory pairs $(\mathbf{y}_w, \mathbf{y}_l)$ from dataset $\mathcal{D}$. TDPO optimizes model performance by enlarging the reward gap between preferred and rejected outputs while penalizing divergence from a reference model, guiding the planner toward adaptive multi-step visual exploration.

Together, the two stages define an efficient and effective training methodology for long-video reasoning, which offers a general recipe for teaching models to reason with lightweight traces, where imitation provides structure and preference optimization provides alignment.

## 4 EXPERIMENT

### 4.1 SETTINGS

**Baselines.** We compare VideoExplorer against both state-of-the-art VLMs and agentic frameworks with retrieval augmentation. For VLMs methods, we compare **GPT-4o** OpenAI (2024) and **Gemini-1.5-pro** Team and etc (2024). For agentic frameworks, we consider: (1). **Vanilla**, which feeds

Table 1: Main experimental results on long video benchmarks. To ensure a fair comparison of reasoning frameworks, all baselines utilize the same visual encoder, compatible with both the Qwen-2.5-VL-7B and 32B models. We report results using each method's fine-tuned LLM where available; otherwise, we use the zero-shot Qwen2.5-32B model.

| Model | LLM Params | VLM Params | Long Video Understanding | | Reasoning | Avg |
|---|---|---|---|---|---|---|
| | | | LVBench | MLVU | MH-NIAH | |
| *Proprietary VLMs* | | | | | | |
| GPT-4o | - | - | 48.9 | 54.9 | - | - |
| Gemini-1.5-pro | - | - | 33.1 | 53.8 | - | - |
| *Direct Reasoning VLMs* | | | | | | |
| Video-R1 | - | 7B | 38.3 | 46.2 | 42.0 | 42.2 |
| *Agentic Frameworks (Qwen2.5VL-7B)* | | | | | | |
| Vanilla | - | 7B | 34.5 | 36.1 | 39.0 | 36.5 |
| VideoAgent | 32B | 7B | 34.9 | 38.6 | 43.0 | 38.8 |
| VideoTree | 32B | 7B | 30.3 | 38.9 | 41.1 | 36.8 |
| VideoRAG | - | 7B | 36.2 | 37.8 | 39.4 | 37.8 |
| Ego-R1 | 3B | 7B | 39.6 | 40.9 | 42.3 | 41.0 |
| **VideoExplorer** | 3B | 7B | 42.0 | 46.3 | 39.9 | 42.7 |
| **VideoExplorer** | 7B | 7B | **50.6** | **55.4** | **49.1** | **51.7** |
| *Agentic Frameworks (Qwen2.5VL-32B)* | | | | | | |
| Vanilla | - | 32B | 34.9 | 41.8 | 39.5 | 38.7 |
| VideoAgent | 32B | 32B | 35.8 | 42.7 | 43.7 | 40.7 |
| VideoTree | 32B | 32B | 32.4 | 39.7 | 41.4 | 37.8 |
| VideoRAG | - | 32B | 37.1 | 42.7 | 39.8 | 39.9 |
| Ego-R1 | 3B | 32B | 41.6 | 43.0 | 42.6 | 42.4 |
| **VideoExplorer** | 3B | 32B | 43.1 | 49.0 | 40.7 | 44.3 |
| **VideoExplorer** | 7B | 32B | **51.4** | **58.6** | **53.4** | **54.5** |

densely sampled frames at 1 fps to the VLM to directly generate answers. (2). **VideoAgent** Wang et al. (2024b), which iteratively locates the informative segments with pre-calculated captions. (3). **VideoTree** Wang et al. (2025), which builds a query-adaptive and hierarchical video representation for LLM reasoning. (4). **VideoRAG** Luo et al. (2024), which employs visually-aligned auxiliary texts to help facilitate cross-modality alignment while providing additional information beyond the visual content. (5). **Ego-R1** Tian et al. (2025) employs a reinforcement learning-trained agent that navigates a hierarchical video index to execute a structured chain-of-thought-for-tool process. We evaluated all models with two different visual modules: Qwen2.5VL-7B and 32B. We preferentially used each model's official weights. For models without publicly available weights, we substituted Qwen2.5-32B and evaluated it in a zero-shot manner. We conducted all analyses using the 7B parameter versions of the LLM and the visual module. Full implementation details for all baselines and VideoExplorer are provided in Appendix B.

**Dataset.** We evaluate our method on two categories of benchmarks: (1) *Long Video Understanding* on LVBench Wang et al. (2024c) and MLVU (test set) Zhou et al. (2024), (2) *Multi-hop Reasoning* on MH-NIAH Li et al. (2025) (max frames is set to 10,000). We report accuracy for all benchmarks. Besides, we assessed temporal grounding effectiveness using the IoU@0.1 metric, which compares the framework-grounded temporal segments to the golden temporal intervals provided by LVBench.

## 4.2 MAIN EXPERIMENT

In Table 1 and Table 2, we present the main experimental results and temporal grounding results on the LVBench, from which we have several key findings: (1) Search-enhanced reasoning methods (e.g. VideoAgent, Ego-R1 and VideoExplorer) show particular strength on multi-hop reasoning tasks such as MH-NIAH. Their ability to iteratively decompose queries, formulate subgoals, and retrieve contextually relevant information mirrors human-like problem-solving behavior and leads to consistently stronger results compared to static, one-shot retrieval pipelines such as VideoRAG. VideoExplorer further benefits from reward-guided reasoning, which encourages the generation of effective sub-queries and efficient trajectories. (2) VideoExplorer significantly outperforms all base-

Table 2: Evaluating Temporal Grounding Accuracy and Question Answering Accuracy on the LVBench Dataset.

| Model | VLM | IoU@0.1 | Accuracy |
|---|---|---|---|
| VideoAgent | 7B | 16.7 | 35.9 |
| VideoRAG | 7B | 14.5 | 36.2 |
| Ego-R1 | 7B | 19.6 | 39.6 |
| **VideoExplorer** | 7B | **27.8** | **50.6** |

Table 3: Ablation Study. Orange, purple and green rows indicate ablations on framework effectiveness, training strategy, and dataset effectiveness.

| Settings | MLVU | MH-NIAH |
|---|---|---|
| **VideoExplorer** | **55.4** | **49.1** |
| – w/o Decoupled TG | 53.0 | 47.3 |
| – w/o TDPO | 52.1 | 47.0 |
| – w/o Difficulty Sampling | 53.3 | 47.6 |

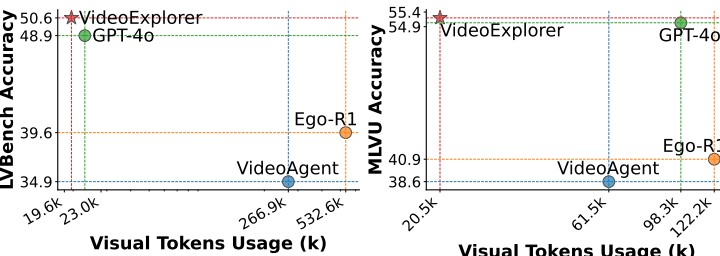

Figure 4: Visual Token Usage Comparison.

lines across all the benchmarks, demonstrating strong robustness and generalizability. This reveals the effectiveness of reasoning with on-demand temporal grounding and visual perception, leading to robust performance across both general long video understanding benchmarks and complex reasoning tasks such as multi-hop reasoning and tutorial reasoning. (3) Compared with retrieval-augmented agentic frameworks, VideoExplorer achieves higher temporal grounding accuracy. Existing frameworks struggle to precisely localize relevant segments, leading to lower QA accuracy. In contrast, VideoExplorer decomposes questions into reasoning steps, aligns them with relevant video segments, and answers based on accurately grounded evidence. As shown in Table 2, prior agentic frameworks fail to reliably identify key segments, resulting in lower overall accuracy, while our method significantly outperforms baselines in IoU@0.1 and QA accuracy. This confirms our hypothesis that adaptive, reasoning-aware temporal grounding is crucial for long-video understanding. VideoExplorer's principled integration of temporal grounding into the reasoning loop, guided by downstream rewards, allows it to dynamically navigate complex visual spaces more effectively than single-turn or static strategies.

## 4.3 ABLATION STUDY

We evaluate the impact of VideoExplorer's components through ablation experiments (Table 3). (1) **Training Strategies:** Removing TDPO causes a notable drop of 3.3% on MLVU and 2.1% on MH-NIAH, showing that it is crucial for stable optimization. By enlarging the reward gap between preferred and rejected outputs based on environment feedback, TDPO encourages the planner to avoid shallow reasoning and instead pursue adaptive, multi-step exploration, ultimately improving the reliability of generated trajectories. (2) **Framework Design:** Decoupling temporal grounding from reasoning reduces the planner's burden and prevents premature answering. Without the grounding agent, performance decreases by 2.4% and 1.8%, as the planner struggles to gather sufficient evidence before producing an answer. This highlights that explicit grounding enables the planner to focus on higher-level reasoning while relying on the grounder to supply relevant temporal spans. (3) **Dataset Effectiveness:** Using only uniformly sampled data leads to 2.1 and 1.5 point drops, confirming that difficulty-adaptive sampling plays an important role. By exposing the model to harder and more diverse cases, it strengthens the reasoning ability and prevents overfitting to trivial patterns, resulting in better generalization on evaluation benchmarks. (4) **Model Scaling:** Expanding the planner from 3B to 7B yields consistent improvements across both datasets, indicating that VideoExplorer can effectively scale with larger model capacity. This suggests that the modular design aligns well with stronger backbones and provides a clear path for further enhancement. Overall, the ablations demonstrate that each component, including training strategies, framework design, dataset construction, and model scaling, contributes meaningfully to the final performance, and their integration is key to the effectiveness of VideoExplorer.

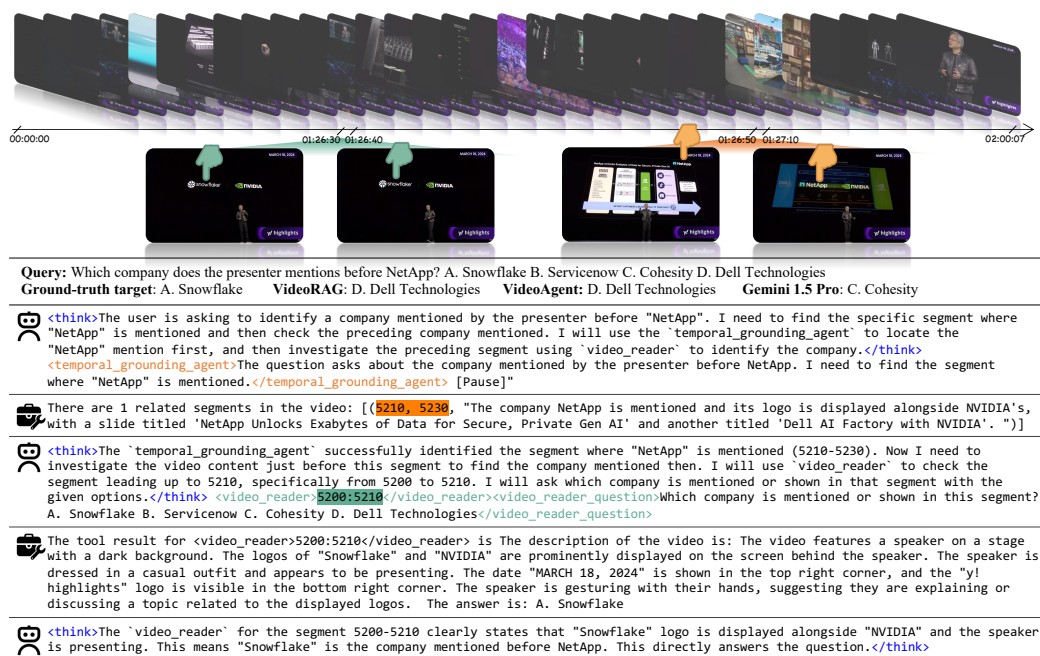

**Query:** Which company does the presenter mentions before NetApp? A. Snowflake B. Servicenow C. Cohesity D. Dell Technologies
**Ground-truth target**: A. Snowflake    **VideoRAG**: D. Dell Technologies    **VideoAgent**: D. Dell Technologies    **Gemini 1.5 Pro**: C. Cohesity

```
<think>The user is asking to identify a company mentioned by the presenter before "NetApp". I need to find the specific segment where
"NetApp" is mentioned and then check the preceding company mentioned. I will use the `temporal_grounding_agent` to locate the
"NetApp" mention first, and then investigate the preceding segment using `video_reader` to identify the company.</think>
<temporal_grounding_agent>The question asks about the company mentioned by the presenter before NetApp. I need to find the segment
where "NetApp" is mentioned.</temporal_grounding_agent> [Pause]
```

```
There are 1 related segments in the video: [(5210, 5230, "The company NetApp is mentioned and its logo is displayed alongside NVIDIA's,
with a slide titled 'NetApp Unlocks Exabytes of Data for Secure, Private Gen AI' and another titled 'Dell AI Factory with NVIDIA. ")]
```

```
<think>The `temporal_grounding_agent` successfully identified the segment where "NetApp" is mentioned (5210-5230). Now I need to
investigate the video content just before this segment to find the company mentioned then. I will use `video_reader` to check the
segment leading up to 5210, specifically from 5200 to 5210. I will ask which company is mentioned or shown in that segment with the
given options.</think> <video_reader>5200:5210</video_reader><video_reader_question>Which company is mentioned or shown in this segment?
A. Snowflake B. Servicenow C. Cohesity D. Dell Technologies</video_reader_question>
```

```
The tool result for <video_reader>5200:5210</video_reader> is The description of the video is: The video features a speaker on a stage
with a dark background. The logos of "Snowflake" and "NVIDIA" are prominently displayed on the screen behind the speaker. The speaker is
dressed in a casual outfit and appears to be presenting. The date "MARCH 18, 2024" is shown in the top right corner, and the "y!
highlights" logo is visible in the bottom right corner. The speaker is gesturing with their hands, suggesting they are explaining or
discussing a topic related to the displayed logos.  The answer is: A. Snowflake
```

```
<think>The `video_reader` for the segment 5200-5210 clearly states that "Snowflake" logo is displayed alongside "NVIDIA" and the speaker
is presenting. This means "Snowflake" is the company mentioned before NetApp. This directly answers the question.</think>
<answer>A</answer>   ← VideoExplorer
```

Figure 5: Case Study of VideoExplorer on LVBench. VideoExplorer correctly decomposes the task, and progressively identifies the relevant video segment through iterative reasoning with temporal grounding. It then performs fine-grained dense perception on this segment to deliver an accurate answer. In contrast, baseline methods all fail on this type of multi-hop fine-grained reasoning task.

### 4.4 TOKEN EFFICIENCY

We compare total visual token usage of different methods in Figure 4. Unlike methods such as VideoAgent and Ego-R1, which indiscriminately process all video segments and thus incur heavy token consumption, VideoExplorer performs dynamic reasoning by selectively focusing only on on-demand task-relevant segments. Despite using far fewer visual tokens, VideoExplorer achieves significantly better performance. This demonstrates its ability to progressively reason and precisely localize meaningful video content, leading to both higher efficiency and stronger effectiveness.

### 4.5 CASE STUDY

We illustrate VideoExplorer's reasoning on an LVBench example in Figure 5. The question, "Which company does the presenter mention before NetApp?", requires multi-hop reasoning: locating NetApp and then tracing backward. Gemini-1.5-pro fails due to downsampling, and traditional RAG methods miss evidence since one-pass retrieval cannot support multi-hop reasoning. In contrast, VideoExplorer decomposes the task, explores iteratively, and finds the correct answer, demonstrating reasoning through dynamic exploration. Additional cases and failure analyses are in Appendix C.

## 5 CONCLUSION

In this work, we introduce VideoExplorer, a novel agentic framework for long-video understanding that follows the principle of "thinking with video". VideoExplorer decomposes complex tasks into sub-questions, adaptively grounds relevant temporal spans, and dynamically adjusts perceptual granularity to enable faithful and interpretable reasoning. This design overcomes the limitations of approaches that rely solely on downsampled frames or static video representations. To fully unlock its capabilities, we curate a specialized dataset and develop a two-stage training pipeline tailored for reasoning-intensive video tasks. Extensive experiments on popular long-video understanding and reasoning benchmarks show that VideoExplorer significantly outperforms existing methods, demonstrating its robustness, adaptability, and efficiency in agentic video understanding.

Ethics Statement

Our dataset builds on the open-source VideoMarathon Lin et al. (2025), which has undergone ethical review by its creators and is released under the Apache License 2.0. We did not modify the original videos; instead, we added textual annotations (reasoning trajectories) to construct the dataset. Our models are initialized from the open-source Qwen-2.5-3B and Qwen-2.5-7B, trained solely on this dataset without harmful content. If any ethical concerns arise, please contact us for resolution.

Reproducibility Statement

To comply with anonymity requirements, we provide the two-stage training and evaluation code in an anonymous repository [2]. Upon acceptance, we will release the full codebase, model checkpoints, and training dataset publicly.

---

[2]https://anonymous.4open.science/r/Video-DeepResearch-CCEF

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

## A USE OF LLMS

The entire research project, including the formulation of the core idea, all experimental work was carried out without LLM assistance.

## B IMPLEMENTATION DETAILS

Our experiments are based on the open-source implementations of most baselines. As VideoAgent's code is not publicly available, we replicated the method based on its paper. To ensure the fairness and rationality of our evaluation, we implement the following controls: (1). The maximum input length of the VLM is set to 32 frames. For Vanilla and VideoRAG, each video is uniformly downsampled to 32 frames as input to the VLM; for Ego-R1 and VideoExplorer, each VLM call to the VLM does not exceed 32 frames in length. (2). For all benchmarks, all baselines rely solely on visual information to answer questions; audio is not provided to prevent unfairness due to information asymmetry.

For VideoAgent and VideoTree, raw videos are segmented into 4s video segments and dense captioned by the VLM. For the Ego-R1 baseline, we adopted the implementation from its official code repository, which builds a hierarchical index using both 10-second and 600-second segments. Both Qwen2.5-VL-7B and Qwen2.5-VL-32B are deployed using the vLLM inference framework to accelerate the inference process. All experiments are conducted on 1 node of 8*80G H100 gpu.

For VideoExplorer, we initialize both planner and temporal grounder with Qwen2.5-7B . We use LanguageBind Zhu et al. (2023) as the video clip embedder. For textual query, we use Language-Bind Zhu et al. (2023) to calculate the text embedding, and calculate similarity with the video clip embeddings. For multimodal query, we average the textual embedding and query image embeding to get the multimodal embedding, and calculate similarity with the video clip embeddings. We set top-k to 10 to balance the accuracy and efficiency. The maximum turns of planner is 20.

## C SUPPLEMENTARY CASE STUDIES AND FAILURE ANALYSIS

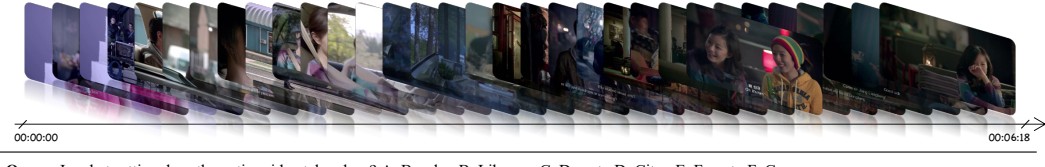

00:00:00                      00:06:18

**Query:** In what setting does the entire video take place? A. Beach   B. Library   C. Desert   D. City   E. Forest   F. Campus
**Ground-truth target**: D    **VideoAgent**: F    **Gemini 1.5 Pro**: D

🤖 `<think>`The question asks about the overall setting of the entire video. This is a holistic question that the `video_browser` agent is best suited to answer. I will use `video_browser` to get a general idea of the video's setting, and then use `temporal_grounding_agent` and `video_reader` to confirm it if necessary.`</think><video_browser>`In what setting does the entire video take place? A. Beach B. Library C. Desert D. City E. Forest F. Campus`</video_browser>`

🧰 The tool results for `<video_browser>`In what setting does the entire video take place? A. Beach B. Library C. Desert D. City E. Forest F. Campus`</video_browser>` is:The video features scenes that include a car interior, a street with a yellow bus, and a lively street market, which are all typical of a city environment. Therefore, the correct answer is D. City.

🤖 `<think>`The `video_browser` agent has provided a clear answer, stating that the video's setting is a city based on the scenes of a car interior, a street with a yellow bus, and a lively street market. This seems to be a sufficient answer for the holistic question. I will proceed with this answer.`</think><answer>`D`</answer>`" ← **VideoExplorer**

Figure 6: Case Study of Holistic Task on MLVU.

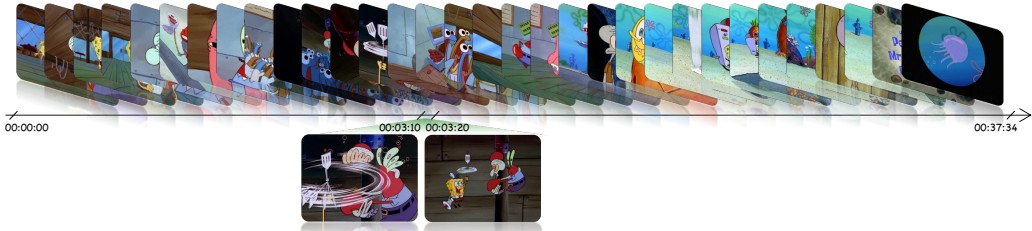

**Query:** Which of the following options correctly matches the sequence of actions as they actually appear in the video?
A. jetskiing --> shredding paper --> paragliding --> abseiling
B. abseiling --> jetskiing --> paragliding --> shredding paper
C. paragliding --> shredding paper --> abseiling --> jetskiing
D. abseiling --> shredding paper --> jetskiing --> paragliding
E. jetskiing --> paragliding --> abseiling --> shredding paper
F. shredding paper --> jetskiing --> abseiling --> paragliding
**Ground-truth target**: F    **VideoAgent**: F.    **Gemini 1.5 Pro**: C.

`<think>` The user is asking to identify the correct sequence of four actions as they appear in the video. I need to find the temporal order of "jetskiing", "shredding paper", "paragliding", and "abseiling". I will use the temporal grounding agent to locate each of these actions and then determine their order.`</think>` `<temporal_grounding_agent>`Locate the segments for "jetskiing", "shredding paper", "paragliding", and "abseiling". `</temporal_grounding_agent>`

There are 4 related segments in the video: [(200, 220, 'Jetskiing'), (40, 50, 'Shredding paper'), (600, 610, 'Abseiling'), (630, 640, 'Paragliding')]

`<think>`I have received the timestamps for all four actions: - Jetskiing: 200-220 - Shredding paper: 40-50 - Abseiling: 600-610 - Paragliding: 630-640  Now I need to arrange them in chronological order based on their start times. 1. Shredding paper (40-50) 2. Jetskiing (200-220) 3. Abseiling (600-610) 4. Paragliding (630-640)  Let's check the options. A. jetskiing --> shredding paper --> paragliding --> abseiling (Incorrect) B. abseiling --> jetskiing --> paragliding --> shredding paper (Incorrect) C. paragliding --> shredding paper --> abseiling --> jetskiing (Incorrect) D. abseiling --> shredding paper --> jetskiing --> paragliding (Incorrect) E. jetskiing --> paragliding --> abseiling --> shredding paper (Incorrect) F. shredding paper --> jetskiing --> abseiling --> paragliding (Correct based on 40-50, 200-220, 600-610, 630-640)  The sequence derived from the timestamps matches option F.`</think>` `<answer>`F`</answer>` ← **VideoExplorer**

Figure 7: Case Study of Multi-detail Task on MLVU.

**Query:** What did the yellow sponge use to fly into the sky?    A. Wings    B. Spoon    C. Boat    D. Pot Lid    E. Bamboo Dragonfly    F. Shovel
**Ground-truth target**: F      **VideoAgent**: A      **Gemini 1.5 Pro**: A

`<think>`The user is asking a question about what the yellow sponge used to fly. I need to first locate the segment where the yellow sponge flies into the sky. Then, I will examine that segment to determine what it used for lift-off. I will use the temporal grounding agent to find the relevant segment.`</think>``<temporal_grounding_agent>`The yellow sponge uses something to fly into the sky. What did it use? A. Wings B. Spoon C. Boat D. Pot Lid E. Bamboo Dragonfly F. Shovel`</temporal_grounding_agent>`

There are 4 related segments in the video: [(190, 200, 'The yellow sponge flies using a spatula and megaphone. '), (210, 220, 'The yellow sponge flies using a spatula as a propeller. '), (390, 400, 'The yellow sponge flies using a large, white, rectangular device with a red propeller at the back. '), (420, 430, 'The yellow sponge flies using a modified boat and a mechanical device attached to it. ')]

`<think>`The temporal grounding agent found four segments where the yellow sponge is flying and being propelled by different means. The question asks what it used to "fly into the sky", implying the initial means of ascent. I need to check each segment to confirm if it's the "initial" use. I will start by examining the earliest segment provided by the temporal grounding agent that mentions flight. The earliest relevant segment is (190, 200). I will ask a specific question about what is used for flying in that segment, including all the options to help the video reader. I will then proceed to other segments if needed.`</think>` `<video_reader>`190:200`</video_reader>``<video_reader_question>`What did the yellow sponge use to fly or ascend in this segment? A. Wings B. Spoon C. Boat D. Pot Lid E. Bamboo Dragonfly F. Shovel`</video_reader_question>`

The tool result for `<video_reader>`190:200`</video_reader>` is The description of the video is: In this segment from an animated show, a yellow sponge character is seen holding a spoon and a pot lid. The sponge character is standing on a wooden surface, and there are other characters in the background. The sponge character then uses the pot lid to fly or ascend into the air, while the other characters watch. The environment appears to be indoors, possibly on a ship or a boat, given the wooden planks and the presence of other characters in the background. The sponge character's expression is one of excitement and determination as it uses the pot lid to fly.  The answer is: D. Pot Lid

`<think>`The video reader for the segment 190:200 explicitly states that "The sponge character then uses the pot lid to fly or ascend into the air". This directly answers the question "What did the yellow sponge use to fly into the sky?". Option D, "Pot Lid", matches this description. This seems to be the correct answer. I should confirm if this is the *initial* use if the question implies that. However, since the question is about "What did the yellow sponge use to fly into the sky?", and the video reader clearly states the pot lid is used for flight, this seems to be the correct answer.`</think>` `<answer>`D`</answer>` ← **VideoExplorer**

Figure 8: Case study of a failure on MLVU. The planner correctly identifies the temporal interval, but the VLM fails due to semantic hallucination, leading to wrong answer.

