# OpenReview forum: "VideoExplorer: Boosting Long Video Understanding with Dynamic Temporal Grounding"
_ICLR.cc/2026/Conference — Submitted to ICLR 2026_

### Official Review · Reviewer_HMvR · 2025-10-31

**Soundness:** 2
**Presentation:** 3
**Contribution:** 2
**Rating:** 4
**Confidence:** 3

**Summary:**

This paper presents VideoExplorer, an agentic framework for long-video reasoning that integrates a planner, a decoupled temporal grounder, and a scalable visual perception module under the principle of  “thinking with video.”  The method decomposes complex tasks into sub-queries, grounds them temporally, and iteratively retrieves relevant video segments. To enable training, the authors construct a reasoning-centric dataset with difficulty-adaptive sampling and design a two-stage optimization pipeline—structured imitation (SFT) followed by trajectory-level preference optimization (TDPO)—to teach multi-step reasoning. Experiments across multiple long-video benchmarks demonstrate improved performance.

**Strengths:**

The paper is clearly motivated by the challenges of long-video reasoning and proposes a solution that integrates planning, temporal grounding, and reasoning. The concept of “thinking with video” offers an intuitive and coherent metaphor that connects cognitive reasoning and perception.

The paper reports results on multiple benchmarks such as LVBench, MLVU, MH-NIAH to demonstrate its performance.

The decoupled planner–grounder structure and dynamic temporal scaling allow for interpretable, step-by-step reasoning. This may benefit future extensions or real-world agentic systems.

The reasoning-centric dataset and two-stage optimization (SFT + TDPO) are clearly presented, and the difficulty-adaptive sampling strategy is reasonable to curate non-trivial supervision signals.

Overall, the paper is clearly written and provides quantitative results to demonstrate the effectiveness of the proposed method.

**Weaknesses:**

The novelty is incremental.  The key idea—iterative query decomposition, retrieval-based reasoning, and adaptive temporal localization—has been extensively explored in VideoAgent (Wang et al., 2024b), Ego-R1 (Tian et al., 2025), and VideoTree (Wang et al., 2025). The proposed concept of “thinking with video” largely parallels prior agentic frameworks with retrieval augmentation, which also decompose a global query into multiple sub-queries to iteratively access contextual evidence. The only difference is whether the clues are retrieved from pre-generated textual contexts (e.g., dense captions, object trajectories, audio transitions) or directly from raw video. The “decoupled temporal grounding” seems more like a system-level reorganization than a fundamentally new algorithmic or theoretical contribution.

Lack of quantitative analysis for efficiency and grounding mechanism. In lines 234–236, the authors claim that the temporal grounder reduces reasoning complexity. However, no quantitative analysis or efficiency benchmark is provided—only a qualitative statement (“reduces redundant searches”). It remains unclear how much retrieval complexity is actually reduced, or whether the proposed method is more efficient than baselines such as VideoRAG or VideoTree. Moreover, the temporal grounding process (“construction, retrieval, verification, summarization”) is never formally defined, ablated, or quantified, leaving the core mechanism somewhat ambiguous.

Potential dataset bias and self-bootstrapping issue. The difficulty-adaptive sampling strategy, while intuitively reasonable, lacks empirical validation for its generality and fairness. Since the reasoning trajectories are generated using the authors’ own VideoExplorer model and subsequently used for model training, the dataset construction may introduce self-bootstrapping bias, potentially inflating performance due to data-model coupling. There is no discussion on how this bias is mitigated or evaluated.

TDPO is under-specified and not clearly novel. The proposed Trajectory-level Direct Preference Optimization (TDPO) appears to be a straightforward adaptation of existing DPO-style alignment methods to multi-step trajectories. The paper does not describe how preference pairs are collected, how trajectory-level rewards are computed, or how TDPO differs substantially from prior RLHF/DPO variants. Without such clarification, the contribution of TDPO remains vague and incremental.

Missing in-depth analysis. While quantitative results show consistent improvements, the paper does not analyze why the method works—e.g., whether gains arise from improved temporal localization, better reasoning decomposition, or dataset curation. There is also no qualitative or failure analysis beyond a single example, which weakens the interpretability and contribution of the approach.

Inconsistent performance under smaller LLM backbones. As shown in Table 1, when the LLM parameter size is 3B, the proposed VideoExplorer performs significantly worse in reasoning compared to both Agentic Frameworks (Qwen2.5-VL-7B) and (Qwen2.5-VL-32B). The paper did not provide an explanation on that, which raises concerns about the method’s robustness and scalability across different model sizes.

**Questions:**

See Weaknesses section above.

---

> ### Author Response · Authors · 2025-11-20
> **Response (Part 1)**
>
> ### **1. Novelty Compared With Prior Agentic Systems**
>
> #### **Review W1:**
>
> The contribution appears incremental; query decomposition, retrieval-based reasoning, and temporal localization have been explored in VideoAgent, VideoTree, and Ego-R1. The proposed “decoupled temporal grounding” may be a system-level rearrangement rather than an algorithmic contribution.
>
> #### **Response:**
>
> Thank you for the insightful comment. While our framework builds upon the general paradigm of iterative decomposition and retrieval, VideoExplorer introduces a **learnable and training-enhanced architecture** that provides **clear advantages in long-horizon temporal reasoning** over prompting-based agents.
>
> 1. **RL-optimized multi-step reasoning.**
>    Prior agents (VideoAgents/VideoTree/VideoRAG) rely on prompting general-purpose LLMs, which are not explicitly optimized for temporal chaining. Our trajectory-level reinforcement learning (TDPO) aligns full reasoning chains with final correctness, leading to **more coherent step decomposition and more stable timestamp transitions**. On MH-NIAH, VideoExplorer achieves **53.4%**, outperforming the next best agent (**43.7%**).
> 2. **Structurally efficient decoupled grounding.**
>    Concurrent end-to-end retrieval–reasoning loops (e.g., Ego-R1) increase token cost and introduce unstable cycles. Our **decoupled grounder** consolidates low-level retrieval operations into a single call, reducing context expansion and improving reasoning correctness.
>    - Table 1 (of the paper): **+20.7%** over Ego-R1
>    - Table 3 (of the paper): **+4.3%** (MLVU) and **+3.7%** (MH-NIAH) over end-to-end grounding
>
> These results demonstrate that VideoExplorer is **not merely a reorganization** of prior agents, but a **learnable and structurally efficient framework** that consistently improves long-video reasoning in our experiments.
>
> ------
>
> ### **2. Efficiency Quantification & Grounding Mechanism**
>
> #### **Review W2:**
>
> The claim that the temporal grounder reduces reasoning complexity lacks quantitative analysis. The grounding process (“construction, retrieval, verification, summarization”) is not clearly defined or ablated.
>
> #### **Response:**
>
> 1. **Quantifying reasoning complexity reduction.**
>    The decoupled grounder shortens planner trajectories by encapsulating low-level retrieval steps. On LVBench:
>
> **Table K:  Token Comsumption Comparison of Different Frameworks**
>
>    | Method | Token Count | Temporal IoU | QA Acc.  |
>    | -- | -- | -- | -- |
>    | No grounder | 8303 | 25.3  | 47.5 |
>    | Ours (decoupled) | 2587 (**↓68.8%**) | **27.8**  | **50.6** |
>
>    The 68.8% reduction confirms a substantial decrease in reasoning complexity. Furthermore, this structural simplification also improves both grounding and QA performance.
>
> 2. **Formalized grounding process.**
>
>    Given any information request, the temporal grounder performs a **four-step** process learned via SFT:
>
>    1. **Sub-query Construction:** Generate refined search queries to improve retrieval coverage and precision [1].
>    2. **Retrieval:** Embed sub-queries and retrieve the top-k candidate segments from the video index.
>    3. **Verification:** Use VLM-based checks to filter out irrelevant or noisy segments.
>    4. **Summarization:** Aggregate verified segments into a concise grounding output for the planner.
>
>    We will clarify this process in the final paper.
>
> 3. **Ablation.**
>    We ablate the verification and summarization steps (construction and retrieval are essential and cannot be removed). The result is as follows:
>
>    | Method  | Temporal IoU | QA Acc.  |
>    | --- | -- | -- |
>    | Full model       | **27.8**     | **50.6** |
>    | w/o Verification | 22.4         | 45.2     |
>
> ​		This experiment shows verification is essential for both precise grounding and downstream QA.
>
> Together, these results confirm that temporal grounding **both reduces reasoning complexity and improves accuracy**.
>
> [1] H. K. Azad and A. Deepak, “Query expansion techniques for information retrieval: A survey,” Inf. Process. Manag., vol. 56, no. 5, pp. 1698–1735, 2019.
>
> ------
>
> ### **3. Dataset Bias & Difficulty-Adaptive Sampling**
>
> #### **Review W3:**
>
> Difficulty-adaptive sampling may introduce model-specific bias, since trajectories are generated using VideoExplorer.
>
> #### **Response:**
>
> To assess generality, we fine-tuned **Qwen2.5-VL**, a model independent of VideoExplorer, on datasets of equal size:
>
> **Table L:  Dataset Generality on General VLMs**
>
> | Training Data  | LVBench  | MLVU     |
> | --- | --- | --- |
> | Random Sampling  | 34.7   | 36.3     |
> | Difficulty-Adaptive Sampling | **36.9** | **38.4** |
>
> Consistent gains on an external VLM indicate that the sampling strategy **improves general reasoning quality** rather than overfitting to VideoExplorer-generated trajectories. We will include this analysis in the revision.

---

> ### Author Response · Authors · 2025-11-20
> **Response (Part 2)**
>
> ### **4. TDPO Specification and Distinction From Prior DPO**
>
> #### **Review W4:**
>
> TDPO appears to be a straightforward extension of DPO; preference construction and reward definition are unclear.
>
> #### **Response:**
>
> 1. **Preference pair construction:**
>    After SFT warm-up, we roll out 4 trajectories and form preference pairs from final correctness:
>
>    - positive: trajectory with correct final answer
>    - negative: trajectory with incorrect final answer
> 2. **Trajectory-level optimization:**
>
> The optimization objective of TDPO is defined as follows.
>
>    $$
>    L_{\text{TDPO}}(\pi_\theta; \pi_{\text{ref}}) = E_{(x,y_w,y_l) \sim D} \left[ \log \sigma \left( \beta \left( \log \frac{\pi_\theta(y_w|x)}{\pi_{\text{ref}}(y_w|x)} - \log \frac{\pi_\theta(y_l|x)}{\pi_{\text{ref}}(y_l|x)} \right) \right) \right]
>    $$
>
>    Key mechanism:
>
>    - Each trajectory probability is computed as
>      $$
>      \pi(y \mid x) = \prod_{t=1}^{T} \pi(a_t \mid x, y_{<t})
>      $$
>
>    - Tool call results are masked to ensure the model learns **reasoning and decision-making**, not the tool output.
>
> Thus, the reward is implicitly encoded by whether a trajectory successfully completes the entire multi-step reasoning chain, directly reinforcing the model to execute the full reasoning process more reliably.
>
> 3. **Difference from DPO/RLHF:**
>
>  - Standard DPO aligns single-hop outputs and cannot enforce coherence over long tool chains. TDPO instead aligns entire multi-step trajectories, supplying supervision at the right granularity for long-video reasoning.
>
> - Online RL is unstable and costly for tool-based video agents, whereas TDPO offers a fast, robust, and effective alternative for multi-step reinforcement.
>
> Overall, we provide specific details on preference construction and trajectory-level reward definition, and demonstrate that TDPO enables fast, robust, and effective optimization of long-horizon multi-step video reasoning with tool calling, without relying on single-step DPO or unstable online RL.
>
> ---
>
> ### **5. In-depth Analysis**
>
> #### **Review W5 (Part 1):**
>
> While quantitative results show consistent improvements, the paper does not analyze why the method works—e.g., whether gains arise from improved temporal localization, better reasoning decomposition, or dataset curation.
>
> #### **Response:**
>
> We clarify that the paper **directly analyzes why VideoExplorer gains performance**, isolating the effects of training, architecture, data strategy, and temporal grounding.
>
> **1. Component Contributions Are Clearly Isolated (Table 3)**
>
> Our ablations have quantified each factor:
>
> * **Training Strategy.** Removing TDPO causes a **3.3% / 2.1%** drop on MLVU / MH-NIAH, showing the effectiveness of our proposed reinforcement learning strategy.
> * **Framework Design.** Disabling the decoupled grounder reduces accuracy by **2.4% / 1.8%**, confirming that separating temporal localization from high-level planning yields clearer, more effective reasoning.
> * **Data Strategy.** Replacing difficulty-adaptive sampling with uniform sampling causes a **2.1% / 1.5%** decline, indicating that the sampling strategy boosts reasoning robustness.
>
> These results reveal that the gains come from improvements in reasoning structure, training stability, and data usage.
>
> **2. Better Temporal Localization Boosts Performance (Table 2)**
>
> Table 2 shows that VideoExplorer achieves **higher temporal IoU** than prior methods, indicating more accurate temporal localization, which supports more reliable multi-step reasoning.
>
> Overall, VideoExplorer’s improvements come from the combined effects of **stronger temporal grounding**, **clear modular reasoning**, and **training/data strategies tailored for multi-step inference**.
>
> ---
>
> ### **6. Qualitative and Failure Cases**
>
> #### **Review W5 (Part 2):**
>
> There is also no qualitative or failure analysis beyond a single example, which weakens the interpretability and contribution.
>
> #### **Response:**
>
> We clarify that **our submission already includes three additional qualitative examples and failure cases in Appendix C**, not just the single illustration shown in the main text. These additional analyses highlight the model’s multi-step reasoning and tool-calling behaviors, as well as typical failure modes, thereby **strengthening the interpretability and contribution of the approach** by providing a more complete view of its behavior, strengths, and limitations.
>
> ---
>
> ### **7. Performance With Smaller Backbones**
>
> #### **Review W6:**
>
> The 3B planner performs worse than 7B/32B agents; scalability concerns remain.
>
> #### **Response:**
>
> While the 3B model already surpasses larger zero-shot baselines on several tasks, its limited capacity constrains long-horizon multi-hop reasoning. The same framework applied to a **7B backbone consistently improves performance across all benchmarks**, demonstrating that VideoExplorer’s architecture and training pipeline scale reliably with model size. We will clarify this observation in the revision.

---

> > ### Comment · Reviewer_HMvR · 2025-11-27
> > **Final comments after rebuttal**
> >
> > I appreciate the detailed response by the authors. However, I still believe that the proposed system is incremental as it largely builds on existing agentic and retrieval-based paradigms. Although the authors add RL-based trajectory optimization and a decoupled grounding module, these updates are incremental in my opinion rather than conceptually novel. The overall pipeline remains close to prior work, and the level of innovation does not meet the typical expectations for a machine learning conference like ICLR.
> >
> > I see that Reviewer gPaM and Reviewer F2uk have also pointed out the incremental nature of the proposed system and lack of novelty.

---

> > > ### Author Response · Authors · 2025-11-27
> > >
> > > Thanks for your response. Just to clarify the remaining misunderstanding within the comments.
> > >
> > > - **Technical novelty in optimization method**.  We focus on **effective reinforcement of  agentic video udnerstanding**, while the mentioned related work are merely hand-crafted workflow without proper reinforcement. Our experiment studies in **Table 3** validates the significant improvement from our proposed RL method (TDPO).
> > > - **Technical novelty in information seeking**.  We proposed a novel information
> > >   seeking approach through grounding, which is fundamantally different from traditional video retrievers and proved to be more effective (see results in Table 3 and additional component-level ablations in Table B).
> > >
> > > We kindly ask the reviewers to pay attention to the above misunderstanding in your previous comments, re-consider the original evaluation to our paper.

---

> ### Author Response · Authors · 2025-11-26
>
> Dear `Reviewer HMvR`,
>
> Thank you once again for your constructive feedback. We have carefully addressed all raised concerns. Our revisions include:
>
> - **Clarified Novelty of VideoExplorer.**
>   We clearly distinguish VideoExplorer from prior prompting-based agents by emphasizing its RL-optimized trajectory reasoning and decoupled temporal grounder.
> - **Efficiency Qualification & Grounding Mechanism Demostration.**
>    We provide quantitative evidence (68.8% token reduction with higher IoU/QA) and formally outline the four-step grounding workflow, supported by existing ablation results.
> - **Dataset Bias Check.**
>   We verify that our constructed dataset based on difficulty-adaptive sampling also boosts a general VLM (Qwen2.5-VL), indicating improved general reasoning rather than system-specific bias.
> - **TDPO vs DPO/RLHF.**
>   We specify trajectory-level preference construction and highlight why TDPO is better suited than single-step DPO or unstable online RL for long-horizon tool reasoning.
> - **Source of Improvements.**
>   We explain that the performance gains primarily stem from stronger temporal grounding, stable trajectory-level training, and improved data sampling, as evidenced by our ablation results.
> - **Qualitative & Failure Cases.**
>   We point readers to multiple qualitative examples and failure analyses in the appendix to better illustrate behavior and limitations.
> - **Scaling with Model Size.**
>   We clarify that while the 3B model has capacity limits, the same framework scales effectively with 7B, yielding consistent performance gains.
>
> We kindly hope that these updates and clarifications will encourage you to reconsider your evaluation, as they directly address your constructive feedback. Should you have any additional queries or reservations, please feel free to contact us at any time. We are fully committed to addressing all concerns to your satisfaction.
>
> Best regards,
>
> Authors

---

### Official Review · Reviewer_iHcw · 2025-11-01

**Soundness:** 3
**Presentation:** 2
**Contribution:** 3
**Rating:** 6
**Confidence:** 4

**Summary:**

The authors propose VideoExplore, an agent-based long video reasoning framework, to address the "task-agnostic, static and lossy representation" issues in existing methods. The framework includes a planner to decompose questions into task-relevant sub-questions, a grounding module to localize key timestamps, and the understanding module to collect the answer to each sub-question. The authors also collect a large-scale training dataset for the 2-stage SFT-RL training strategies. The experimental results demonstrate its superior performance over existing methods. However, the writing in the methods section needs improvement for a better reading experience.

**Strengths:**

1. The authors proposed VideoExplore, which is a sophisticated and effective agent-based video reasoning framework for long video understanding and reasoning.
2. To facilitate the 2-stage training, the authors collected a large-scale training dataset (including QA and reasoning chains) based on existing long video datasets and with the help of LLMs.
3. The experiments demonstrate that the proposed methods outperform previous methods by a large margin.
4. Ablation studies demonstrate the effectiveness of the components of the framework, the training strategies, and the training data.
5. The visual token usage also indicates a high computation efficiency compared with other methods.

**Weaknesses:**

Major:
1. The writing in 3.2 is not so clear. In Eq. 4, the definition and format of P_t, and the format of T_t (temporal span), are missing. They can be somehow inferred from Figure 1, but a clear description in 3.2 will help reading. And also, if there is a function of the planner, it will be more complete for the task definition.
2. In line 235, "In the offline stage, videos are segmented into clips ...". What is the specific method to segment these videos? Uniformly? Does this step only relate to semantic matching as described in line 236?
3. With these segments, it looks like only text descriptions and timestamps can help the video understanding. What is the purpose and benefit of introducing multimodal queries?
4. This paragraph, "Decoupled Temporal Grounding Execution." is hard to understand. The function of the planner and the passing-back mechanism (line 241) is not illustrated in Figure 1. A simple "Retrieving-->Verifying-->Summarizing" in Figure 1 looks oversimplified for the complex framework.
5. What is the timestamp representation/encoding in the framework? Is it the frame index or absolute time? It looks like there is a discrepancy in Figure 1 and Figure 5.

Minor:
1. In line 045, "To address to the above problems, ..." --> "To address the above problems, ...".
2. In line 146, "... downsampling the raw video video and ..." --> ""\... downsampling the raw video and ...".

**Questions:**

Please refer to the weakness.

---

> ### Author Response · Authors · 2025-11-20
>
> Thank you for the constructive comments. We have clarified key definitions, improved the description of the grounding mechanism, and corrected inconsistencies and typos. We believe these revisions enhance the clarity of the paper.
>
> ---
>
> ### **1. Clarification of Notation and Definitions in Section 3.2**
>
> #### **Review (Major 1):**
>
> The writing in Section 3.2 is unclear. In Eq. 4, the definitions and formats of $P_t$ and $T_t$ are missing. They can be inferred from Figure 1, but a clear description is needed. In addition, providing the function of the planner would make the task definition more complete.
>
> #### **Response:**
>
> Thank you for highlighting the ambiguity in Section 3.2. We have revised the section to explicitly define the temporal span, the perception output format, and the planner function.
>
> **(1) Video Segment $\tau_t$.**
>
> The video segment is defined as:
>
> $$
> \tau = (s, e),\qquad 0 < s < e < D,
> $$
>
> where $D$ is duration of the long video.
>
> **(2) Grounding output $T_t$**
>
> $$
> T_t = [\tau_{t,1}, \ldots, \tau_{t,N_t}]
>     = \Gamma_{\text{ground}}\bigl(q_t^{\text{sub}} \mid v\bigr),
> $$
>
>
> where $\Gamma_{\text{ground}}$ is the temporal grounder, and $N_t$ is the number of retrieved segments.
>
> **(3) Perception output $ \mathcal{P}_t $.**
>
> The perception module takes a sub-query $q_t^{\text{sub}}$ and timestamp interval $\tau_t$ and produces corresponding textual answer using either the Video Browser or Video Reader:
>
> $$
> P_t
> = (\Phi_{\mathrm{perc}}(q_t^{sub}, \tau_{t,i}))_{i=1}^{N_t}.
> $$
>
> where $\Phi_{\text{perc}}$ is the parameters of video reader/video browser.
>
> Note: Since curly braces cannot be reliably rendered in the OpenReview rebuttal box, we use parentheses to denote sequences for clarity.
>
> **(4) Planner function with unified history $\mathcal{H}_t$.**
> To make the task formulation complete, we define the planner’s input as the entire reasoning history:
>
> $$
> H_t = ( (\ r_i,\ a_i), P_i ))_{i=1}^{t}.
> $$
>
> The planner then generates the next reasoning step and the next action:
>
> $$
> r_{t+1},\ a_{t+1} = \Theta\left(\mathcal{H}_t \right),
> $$
>
> where $\Theta$  is the parameters of the planner.
>
> **(5) Final Answer output.**
>
> $$
> A = \Theta\left(q \mid (H_t)_{t=1}^{T}\right).
> $$
>
> We will update Section 3.2 accordingly to make it more clear.
>
> ---
>
> ### **2. Details of Video Segmentation**
>
> #### **Review (Major 2):**
>
> Line 235 states that videos are segmented into clips. What is the method for segmentation? Uniform segmentation? Is this step only used for semantic matching?
>
> #### **Response:**
>
> Thank you for this question. In our method, videos are uniformly segmented into non-overlapping clips of 10 seconds each to build the search index. As you correctly noted, these segmented clips are used exclusively for the purpose of semantic matching, as mentioned in line 236.
>
> ---
>
> ### **3. Purpose of Multimodal Queries**
>
> #### **Review (Major 3):**
>
> Given that the method uses textual descriptions and timestamps, what is the purpose and benefit of adding multimodal queries?
>
> #### **Response:**
>
> We thank the reviewer for raising this important question. The introduction of multimodal queries is primarily aimed at enhancing the model’s ability to resolve ambiguities that are often present in purely textual descriptions.
>
> As illustrated in Figure 1, a textual query such as “Locate a guy gathered with other players” can be ambiguous, since the term “a guy” may refer to multiple individuals in a given video scene. By incorporating a visual example, such as an image of the specific person referred to as “he”, the model can more precisely identify the target subject, thereby reducing reliance on subjective interpretation of the text alone.
>
> Thus, the key benefit of multimodal queries lies in their capacity to complement textual information with visual evidence, leading to more accurate and robust video segment retrieval. We appreciate the reviewer’s insightful comment and hope this clarification adequately addresses the concern.

---

> > ### Author Response · Authors · 2025-11-20
> >
> > ### **4. Clarifying “Decoupled Temporal Grounding Execution”**
> >
> > #### **Review (Major 4):**
> >
> > The paragraph describing “Decoupled Temporal Grounding Execution” is difficult to understand. The function of the planner and the passing-back mechanism (line 241) is not illustrated in Figure 1. The “Retrieving→Verifying→Summarizing” depiction seems oversimplified.
> >
> > #### **Response:**
> >
> > Thank you for your valuable feedback regarding the description of the "Decoupled Temporal Grounding Execution" and the representation in Figure 1. We appreciate the opportunity to clarify these points in more detail.
> >
> > 1. **Meaning of "Decoupled Temporal Grounding Execution":**
> >    This term refers to the separation (or decoupling) of the temporal grounding process from the main reasoning trace of the planner. In other words, instead of having the planner handle both high-level reasoning and low-level temporal retrieval tasks, we isolate the temporal grounding into a dedicated module.
> >
> > 2. **Function of the Planner and the Passing-Back Mechanism:**
> >    Without decoupling, the planner would be responsible not only for overall reasoning strategy—such as deciding which tools to use and when to produce the final answer—but also for executing the multi-step temporal retrieval process. This process typically involves:
> >
> >    - Constructing a sub-query
> >    - Retrieving relevant temporal segments
> >    - Verifying their relevance
> >    - Summarizing the retrieved information
> >
> >    This four-step procedure is often lengthy and computationally intensive.
> >
> > 3. **How Decoupling Simplifies the Process:**
> >    In our proposed framework, the planner only needs to:
> >
> >    - Formulate high-level reasoning strategies
> >    - Specify its information needs
> >
> >    These information needs are then passed to the dedicated **Temporal Grounder**, which autonomously executes the entire process illustrated in the "Temporal Grounding Execution" part of Figure 1. The Temporal Grounder returns precise and relevant temporal segments back to the planner.
> >
> > 4. **Impact on Workflow Efficiency:**
> >    As a result, what originally required the planner to perform four steps for each retrieval is now reduced to a single step: submitting an information request to the Temporal Grounder. This significantly streamlines the planner’s workload and enhances the overall efficiency of the framework.
> >
> > 5. **Regarding Figure 1:**
> >    This representation is intended to highlight the core stages of the Temporal Grounder’s internal process. In the figure, the planner’s request is "Locate the person who consumed a golden burger," and the returned results are "[[00:34:00-00:34:10], [00:35:50-00:36:00]...]". We will revise the figure in the manuscript to illustrate this process more clearly.
> >
> > We hope this explanation clarifies the intended functionality and the advantages of the decoupled design. Thank you once again for your thoughtful comments.
> >
> > ---
> >
> > ### **5. Timestamp Representation Consistency**
> >
> > #### **Review (Major 5):**
> >
> > What is the timestamp representation—frame index or absolute time? Figures 1 and 5 seem inconsistent.
> >
> > #### **Response:**
> >
> > In our framework, timestamps are represented in **absolute seconds**. The numbers in Figure 5, such as 5210, indeed correspond to 5210 seconds. We apologize for any confusion caused by the discrepancy and will illustrated this in the revised manuscript.
> >
> > ---
> >
> > ### **6. Typos**
> >
> > #### **Review (Minor):**
> >
> > - Line 45: “To address to the above problems…” → “To address the above problems…”
> > - Line 146: “… downsampling the raw video video and …” → “… downsampling the raw video and …”
> >
> > #### **Response:**
> >
> > We appreciate the careful reading and will correct these typos in the revised manuscript.

---

> > > ### Author Response · Authors · 2025-11-26
> > >
> > > Dear `Reviewer iHcw`,
> > >
> > > We sincerely thank the reviewer for the constructive and detailed comments. We have revised the manuscript accordingly and clarify the main updates as follows:
> > >
> > > **1. Notation & Definitions (Major 1).**
> > >  We now explicitly define temporal spans, grounding/perception outputs, and the planner function with unified history. Section 3.2 has been rewritten for clarity.
> > >
> > > **2. Video Segmentation (Major 2).**
> > >  We clarify that videos are uniformly segmented into 10-second clips, used solely for semantic matching in retrieval.
> > >
> > > **3. Purpose of Multimodal Queries (Major 3).**
> > >  We demonstrate that multimodal queries help disambiguate textual descriptions (e.g., identifying a specific person when text alone is insufficient), leading to more accurate grounding.
> > >
> > > **4. Decoupled Temporal Grounding Execution (Major 4).**
> > >  We clarify that the planner focuses on high-level reasoning, while a dedicated grounder handles retrieval, verification, and summarization. This reduces planner burden and improves efficiency. We will revise Figure 1 to better illustrate this workflow.
> > >
> > > **5. Timestamp Representation (Major 5).**
> > >  All timestamps are in absolute seconds. We will ensure consistent notation in Figures 1 and 5.
> > >
> > > **6. Typos (Minor).**
> > >  All noted typos will be corrected.
> > >
> > > Thank you again for your thoughtful comments. We hope these clarifications address your concerns, and we would be happy to provide any further details if needed.
> > >
> > > Best regards,
> > >
> > >  Authors

---

### Official Review · Reviewer_F2uk · 2025-11-05

**Soundness:** 3
**Presentation:** 3
**Contribution:** 2
**Rating:** 4
**Confidence:** 5

**Summary:**

This paper proposes an idea of “Thinking with video.” Analogous to “thinking with images,”  “thinking with video” treats reasoning as a dynamic process of temporally grounded exploration and decomposition. The model iteratively decides what to look for, where to watch, and at what temporal scale, flexibly combining both fine-grained inspection and coarse-grained temporal grounding for long video understanding. The paper also proposes a two-stage training framework with SFT and DPO on accepted temporal grounding trajectories to improve VQA on long video benchmarks and showcase great performance improvement with modest innovation.

**Strengths:**

1. The paper clearly extends the idea of Thinking with Images to the video domain from a temporal perspective.

2. It constructs a reasoning-centric dataset with multi-step reasoning trajectories, supervised for a two-stage optimization pipeline that combines structured imitation learning (SFT) and trajectory-level preference alignment (DPO for video)

3. The work demonstrates strong token efficiency in long-video understanding, outperforming uniform-sampling-based methods.

**Weaknesses:**

1. The proposed components include question decomposition and agent planning, question-aware temporal grounding, agentic iterative reasoning, and temporally grounded preference optimization—have been explored in many existing works such as DrVideo [1], VideoINSTA [2], Traveler [3], TPO [4], and Video-R1 (T-GRPO) [5]. The authors either missed direct relevant comparisons or need to further justify the novelty and necessity of their proposed multi-step temporal grounding approach.

[1] DrVideo: Document Retrieval-Based Long Video Understanding. CVPR 2025.

[2] VideoINSTA: Zero-shot Long Video Understanding via Informative Spatial-Temporal Reasoning with LLMs. EMNLP 2024.

[3] Traveler: A Modular Multi-LLM Agent Framework for Video Question Answering. EMNLP 2024.

[4] Temporal Preference Optimization for Long-Form Video Understanding. arXiv:2501.13919.

[5] Video-R1: Reinforcing Video Reasoning in MLLMs. arXiv:2503.21776.

2. The annotation procedure for multi-step reasoning in the proposed dataset is unclear and less described. The definitions of step-wise reasoning and the quality assurance of VLM-based grounding annotations require more explanation.

3. In Table 2, the QA accuracy shows significant improvement. However, the evaluation of Temporal Grounding Accuracy compared to non-grounding methods (e.g., Ego-R1, VideoAgent) needs additional validation, preferably against other temporal grounding-focused baselines such as [6]

[6] ReVisionLLM: Recursive Vision-Language Model for Temporal Grounding in Hour-Long Videos. CVPR2025.

Formatting Errors:

In Section 2.1, the references following “…leading to the emergence of Multi-modal Large Language Models (MLLMs)” are improperly formatted.

Line 146 contains a duplicate “video”.

**Questions:**

In Section 2.2, regarding the statement: “This paradigm will inevitably lose the rich visual information in original long videos, leading to sub-optimal performance such as in egocentric videos…”
The reviewer finds this argument unconvincing, as captioning from image or video frames does not seem to differ significantly between egocentric and exocentric videos.

---

> ### Author Response · Authors · 2025-11-20
> **Response (Part 1)**
>
> Thank you for the constructive feedback. Below, we **clarify our framework’s novelty**, **explain the annotation process**, **address grounding evaluation**, and fix the noted misunderstandings and formatting issues.
>
> ------
>
> ### **1. Novelty Clarification and Experimental Comparison**
>
> #### **Review W1**
>
> Comparison with DrVideo, VideoINSTA, Traveler, TPO, and Video-R1 is requested.
>
> #### **Response**
>
> Thank you for the constructive comment. Our work targets a different problem from these methods: instead of reasoning over a *fixed context*, we train a model to **dynamically identify information needs and actively retrieve evidence across entire long videos**, which is why we compare primarily with agentic frameworks.
>
> **1. Key distinctions from prior work:**
>
> 1. **Active tool-based reasoning instead of static prompting.**
>    Prior methods rely on prompting over captions or fixed features; they do not learn to plan, decompose, or interact with video tools. VideoExplorer learns planner + grounder end-to-end.
> 2. **Breaks the fixed-context bottleneck.**
>    TPO and Video-R1 operate under 64-frame limits. Our iterative retrieval removes this restriction and supports hour-long videos.
> 3. **Unified optimization of architecture, data, and training.**
>    Unlike works focusing on isolated modules, we jointly optimize decomposition, grounding, and multi-step reasoning via decomposed design, curriculum data, and TDPO.
>
> **2. Experimental comparison (default settings from original papers):**
>
> **Table J: Comparison with Video Reasoning Models**
>
> | Model | LVBench | MLVU |
> | -- | --- | -- |
> | Traveler | 38.3 | 41.1 |
> | VideoINSTA | 33.7 | 39.2 |
> | TPO | 41.5 | 54.1 |
> | Video-R1| 38.0 | 42.0  |
> | **Ours** | **50.6** | **55.4** |
>
> The results show that our model consistently surpasses prompting- and RL-based video reasoning methods on long-video benchmarks. Unlike prompting-based or fixed-context reasoning methods, VideoExplorer continually reinforces its own learning of video understanding through tool use, leading to robust improvements on long-video tasks.
>
> We appreciate the reviewer’s constructive suggestions and will incorporate the corresponding results in the revised version.
>
> ----
>
> ### **2. Clarification of Annotation Procedure**
>
> #### **Review W2**
>
> The annotation procedure for multi-step reasoning is unclear. The definitions of step-wise reasoning and the quality assurance of VLM-based grounding annotations require more explanation.
>
> #### **Response**
>
> Thank you for the comment. We clarify that our annotations **do not use VLM-based grounding**; all reasoning traces are generated by a strong LLM or SFT checkpoint.
>
> 1. **Data Annotation Procedure**
>
>    1. **SFT Dataset:**
>       1. **Preliminary trajectories:**
>          We uniformly sample tasks from VideoMarathon and use DeepSeek-R1 as both planner and grounder. Given the task description, the model interacts with tools until producing a final answer. We retain only trajectories that yield correct answers and use their planner traces and grounding traces to train the planner and temporal grounder.
>       2. **Difficulty-based re-sampling:**
>          To improve data quality and difficulty, we re-sample tasks whose accuracy falls below a threshold θ, regenerating harder examples to enhance robustness.
>    2. **TDPO Dataset:**
>       Using the SFT-enhanced VideoExplorer, we generate 4 rollouts on training data. Trajectories that are partially correct are used to form TDPO training pairs, reinforcing correct multi-step reasoning.
>    3. **Annotation note:**
>       Both SFT and TDPO datasets are generated using a strong LLM or SFT checkpoint as the planner to produce **complete reasoning chains**, filtered by reasoning correctness; **no VLM-based grounding annotations** are used.
>
> 2. **Definition of Step-wise Reasoning**
>
>    As shown in Figure 1, at each step the planner calls the temporal grounder or perception modules (Video Reader / Video Browser) to acquire information until the answer is produced. Formally, given a video–query pair $(v,q)$, VideoExplorer produces a reasoning trajectory $Y_{\text{trace}}$, which consists of the following sequence:
>
>    $Y_{\text{trace}}$ =
>      `<think>` reasoning step `</think>`,
>      `<ground>` temporal span `</ground>`,
>      `<think>` reasoning step `</think>`,
>      `<info>` information patch `</info>`,
>      ...,
>    `<answer>` final answer `</answer>`.
>
>
>    Here, `<think>` denotes planner reasoning, `<ground>` and `<info>` represent outputs from the temporal grounder and perception modules. The planner iteratively decomposes the query, retrieves evidence, and integrates information until generating the `<answer>`.
>
> **In summary**, all annotations are **complete, correctness-verified reasoning chains** produced by a strong LLM or SFT checkpoint, and **do not rely on VLM-based grounding annotations**. We will include a concise description of this step-wise reasoning and data construction process in the revised paper.

---

> > ### Author Response · Authors · 2025-11-20
> > **Response (Part 2)**
> >
> > ### **3. Temporal Grounding Assessment**
> >
> > #### **Review W3:**
> >
> > Request for comparison with grounding-focused baselines such as [6].
> >
> > #### **Response:**
> >
> > ##### **1. Clarification on Temporal Grounding Evaluation**
> >
> > Thank you for the insightful comment. We clarify that **both our method and the compared baselines (e.g., Ego-R1, VideoAgent) are QA-oriented frameworks, not temporal grounding models.** In these systems, **video moments are intermediate reasoning outputs, not final predictions, and are not ranked.**
> >
> > Temporal grounding benchmarks such as [6] require models to output **ranked candidate intervals** for metrics like Recall@1/5. Since neither our method nor these QA-based baselines generate ranked intervals, **direct comparison with grounding-specific systems under such benchmark protocols is not applicable**.
> >
> > Instead, we report *indirect grounding quality* on LVBench (Table 2), which measures alignment between retrieved segments and ground-truth moments during QA reasoning, **providing a fair and consistent evaluation** across QA-focused methods.
> >
> > ##### **2. Extension to Grounding-Oriented Long-Video Tasks**
> >
> > Although our current work targets QA, VideoExplorer can be extended to temporal grounding or moment retrieval tasks with the following adaptations:
> >
> > - **Task-Specific Data:** Curate long-video grounding datasets that provide explicit moment-level supervision.
> > - **Output Format:** Modify the system to output a temporal interval as the final prediction.
> > - **Supervision Signal:** Incorporate interval-level losses (e.g., IoU loss or Smooth L1 loss) to directly optimize temporal boundary prediction.
> >
> > These adaptations are orthogonal to our QA-focused design and can be incorporated when targeting grounding as the primary task. In this paper, our focus is on **efficient multi-step video reasoning for QA**, and our evaluation follows this problem setting.
> >
> > ------
> >
> > ### **4. Clarification of Statement Interpretation**
> >
> > #### **Review Q1:**
> >
> > In Section 2.2, regarding the statement: “This paradigm will inevitably lose the rich visual information in original long videos, leading to sub-optimal performance such as in egocentric videos…” The reviewer finds this argument unconvincing, as captioning from image or video frames does not seem to differ significantly between egocentric and exocentric videos.
> >
> > #### **Response:**
> >
> > Thank you for the reviewer’s careful reading and insightful comment. We would like to clarify that the statement cited in the review does not appear in our paper. Our original wording is:
> >
> > > *“However, these approaches transit videos into task-agnostic textual representations (e.g., object locations, dense captions, audio transitions, etc.), and then leverage powerful LLMs to reason over the textual representations. This paradigm will inevitably lose the rich visual information in original long videos, leading to sub-optimal performance. Additionally, many of these methods are designed for limited scenarios such as egocentric videos (Wang et al., 2024b; Tian et al., 2025), reducing their applicability to more general real-world settings.”*
> >
> > This paragraph makes **two independent points**:
> >  (1) *converting long videos into task-agnostic textual representations inherently discards fine-grained visual and temporal cues*, and
> >  (2) *some existing methods focus on limited domains, such as particular egocentric benchmarks, which affects their general applicability*.
> >
> > We believe the confusion arises from reading these two points as a single joint statement, which implicitly links the information-loss claim specifically to egocentric videos. In our manuscript, the mention of egocentric videos is intended only to illustrate the restricted scope of some prior works, not to attribute information loss to the egocentric setting itself.
> >  We appreciate the reviewer’s comment and will refine this paragraph in the revised version to make this separation explicit.
> >
> > ---
> >
> > ### **5. Formatting Errors**
> >
> > #### **Review:**
> >
> > 1. In Section 2.1, the references following “…leading to the emergence of Multi-modal Large Language Models (MLLMs)” are improperly formatted.
> >
> > 2. Line 146 contains a duplicate “video”.
> >
> > #### **Response:**
> >
> > Thank you for pointing this out. We will fix these formatting errors in our revision.

---

> ### Author Response · Authors · 2025-11-26
>
> Dear `Reviewer F2uk`,
>
> Thank you once again for your detailed and constructive evaluation. We have carefully addressed all raised concerns. Our revisions include:
>
>  - **Additional experimental comparisons with prior prompting-based and fixed-context methods**, supported by additional comparisons with DrVideo, Traveler, VideoINSTA, TPO, and Video-R1. The results will be included in our revision.
>
>  - **A demonstration of correctness-verified annotation pipeline**, where all reasoning traces are produced by a strong LLM or SFT checkpoint without any VLM-based grounding annotations.
>
>  - **An explanation of grounding evaluation**, showing why grounding-specific benchmarks are not directly comparable for QA-oriented agents and why LVBench provides a consistent metric for all compared systems.
>
>  - **A clarification regarding the reviewer’s misinterpreted statement**, explaining that the paragraph in Section 2.2 contains two independent points: information loss from textual conversion and the limited applicability of certain egocentric methods.
>
>  - **Corrections to formatting issues** in Section 2.1 and Line 146.
>
> We sincerely hope that these clarifications and newly added results fully resolve the concerns you identified. If any further questions arise, we are more than happy to provide additional explanations or supplementary evidence.
>
> Thank you again for your thoughtful feedback, which has significantly strengthened the clarity and rigor of our work. We kindly hope that these updates will help you **reconsider your evaluation**.
>
> Best regards,
>
>  Authors

---

### Official Review · Reviewer_BT9o · 2025-11-06

**Soundness:** 3
**Presentation:** 3
**Contribution:** 3
**Rating:** 6
**Confidence:** 3

**Summary:**

This paper addresses long video understanding by introducing VideoExplorer, a framework that iteratively plans, grounds, and reasons over relevant video segments rather than processing a fixed downsampled input. A planner decomposes complex queries into sub-questions, while a temporal grounder retrieves and verifies supporting segments, adjusting frame granularity as needed. The system is trained using a new reasoning-centric dataset with difficulty-adaptive sampling and a two-stage process: supervised fine-tuning followed by trajectory-level preference optimization. This enables VideoExplorer to produce faithful, multi-step reasoning chains. The authors demonstrate state-of-the-art results on several long-video benchmarks, showing improved scalability, accuracy, and interpretability. Overall, the work introduces a dynamic and cognitively inspired paradigm for long video reasoning.

**Strengths:**

- Methodology - VideoExplorer is built from well-justified components—planner, grounder, and a cognitive reasoning loop—and its training combines imitation learning with trajectory-level RL, well-suited for multi-step tasks. The decoupled grounding module effectively handles query ambiguity, and each design choice is validated through experiments, such as performance drops when TDPO is ablated.
- Comprehensive Evaluation - The paper presents a thorough empirical evaluation across multiple benchmarks, where VideoExplorer consistently achieves state-of-the-art results in both accuracy and temporal grounding. Ablation studies, efficiency analysis, and qualitative case studies further support that the performance gains stem from the proposed innovations, reinforcing the method’s robustness and generality.

**Weaknesses:**

- Baselines - The paper evaluates against agentic frameworks and vision-language models that either process uniformly sampled frames or use retrieval-augmented textual representations. However, it does not benchmark against methods that tackle long video understanding by selecting representative keyframes as a compression or reasoning strategy, e.g. [a,b,c,d]. Including such baselines could have strengthened the evaluation by clarifying how much of VideoExplorer’s performance gains come from temporal reasoning and grounding, versus simply improved frame selection. This should also be addressed in related work section.
- Efficiency - The paper shows reduced token processing compared to baselines, but an analysis of runtime or memory overhead would be helpful. Any empirical or theoretical insight into the efficiency limits would strengthen understanding of the method’s practicality.
- Generality - Not really a weakness, but the tasks in the paper are QA-style; it would be interesting to see how well the approach would generalize to other long-video understanding tasks.
- Planner and Grounder Robustness - If the planner formulates off-target sub-questions or the grounder retrieves redundant segments, the model may overlook critical information or waste reasoning steps. The paper does not clearly address how such failures are detected or mitigated, raising concerns about the system’s robustness in complex or ambiguous scenarios.

References:
- [a] M-LLM Based Video Frame Selection for Efficient Video Understanding, CVPR 2025
- [b] VideoEspresso: A Large-Scale Chain-of-Thought Dataset for Fine-Grained Video Reasoning via Core Frame Selection, CVPR 2025
- [c] Vila: Efficient videolanguage alignment for video question answering, ECCV 2024
- [d] Self-Chained Image-Language Model for Video Localization and Question Answering, NeurIPS 2023

**Questions:**

- Have you considered comparing VideoExplorer to keyframe selection methods? This could help isolate how much of the performance gain stems from dynamic temporal grounding versus improved frame selection alone.
- Can you share any measurements or analysis on runtime, memory usage, or compute overhead? How does the multi-turn framework scale with video length in practice?
- Do you believe the VideoExplorer framework could extend to long video understanding tasks other than QA benchmarks? If so, what adaptations might be needed?
- How does the system handle cases where the planner generates sub-optimal queries or the grounder retrieves redundant or irrelevant segments? Are there safeguards to prevent inefficient or incomplete reasoning paths?

**Details Of Ethics Concerns:**

Not Applicable.

---

> ### Author Response · Authors · 2025-11-20
>
> We thank the reviewers for the thoughtful feedback. Below we provide concise clarifications and experiments demonstrating VideoExplorer’s advantages in **accuracy, efficiency, and robustness**.
>
> ---
>
> ### **1. Comparison with Keyframe-based Baselines**
>
> #### **Review W1 & Q1:**
>
> The paper benchmarks against standard agentic and vision-language models but omits comparison with keyframe-based long-video methods, leaving unclear how much performance gains stem from temporal reasoning versus frame selection.
>
> #### **Response:**
> Thank you for the valuable suggestion. We reproduced the open-sourced keyframe-based methods:
>
> **Table F: Comparison with Keyframe-based Methods**
>
> | Methods | LVBench | MLVU |
> | --- | -- | -- |
> | SeVila  | 35.7 | 39.2 |
> | Vila  | 31.3    | 34.1 |
> | VideoExplorer | 50.6    | 55.4 |
> These methods underperform VideoExplorer for two reasons:
>
> 1. These methods **only identify key frames within a limited input window** (SeVila and Vila take 32 and 8 frames, respectively), causing severe information loss in hour-long videos;
> 2. They perform **a single-round key frame selection**, making multi-hop reasoning difficult.
>
> VideoExplorer instead retrieves from the **full video** and **iteratively** decides whether to continue tool use. Thus, our gains stem from full-video reasoning, not improved frame picking. We will include this comparison in the paper.
>
> ---
>
> ### **2. Runtime and Memory Analysis**
>
> #### **Review W2 & Q2:**
>
> The paper shows reduced token processing compared to baselines, but an analysis of runtime or memory overhead would be helpful.
>
> #### **Response:**
>
> Thanks for the constructive advice. We conducted the following experiments for a comprehensive analysis:
>
> **Table G: Inference Efficiency Comparison**
>
> |  | VideoAgent | Ego-R1 | VideoExplorer |
> | -- | --| -- | -- |
> | Inference time per sample (s) | 23.5 | 51.4 | **22.6** |
> | Peak GPU Memory (GB)  | **64.8** | 113.2 | 86.3  |
> | Accuracy | 38.8 | 41.0 | **51.7**  |
>
> VideoExplorer matches VideoAgent in speed while improving accuracy by ~13%. It is also substantially faster and more memory-efficient than Ego-R1, primarily because our decoupled grounding shortens the reasoning chain and avoids the repeated retrieval loops that slow down Ego-R1.
>
> **Overall, VideoExplorer provides the strongest balance of speed, memory, and accuracy.**
>
> ---
>
> ### **3. Generality Beyond QA**
>
> #### **Review W3 & Q3:**
>
> W3: Generality - Not really a weakness, but the tasks in the paper are QA-style; it would be interesting to see how well the approach would generalize to other long-video understanding tasks
>
> Q3: Can VideoExplorer generalize to other long-video tasks, and what adaptations would this require?
>
> #### **Response:**
>
> We appreciate the reviewer’s comments. While our experiments focus on QA tasks, the framework’s design and temporal reasoning capabilities naturally extend to other long-video understanding tasks such as temporal grounding.
>
> **1.  Current grounding behavior.**
> Results in Table 2 of the paper show that VideoExplorer’s retrieved segments align well with ground-truth moments on LVBench (IoU@0.1 = 27.8 vs. baselines 14.5–19.6), indicating reliable temporal localization during reasoning.
>
> **2. Extending to other long-video tasks.**
>
> Taking temporal grounding as an example, VideoExplorer can be adapted by (i) constructing high-quality task-specific datasets, (ii) predicting temporal spans instead of textual answers, and (iii) incorporating alignment losses (e.g., IoU or Smooth L1). We would be glad to see VideoExplorer applied to more long-video tasks.
>
> ---
>
> ### **4. Planner and Grounder Robustness**
>
> #### **Review W4:**
>
> The system’s handling of off-target plans or redundant retrievals is unclear, leaving potential failures in complex scenarios unaddressed.
>
> #### **Response:**
>
> Thank you for raising this concern. We have incorporated several mechanisms to mitigate both off-target planning and redundant grounding:
>
> 1. **TDPO reduces incorrect reasoning, including off-target query decomposition.**
>     It reinforces correct multi-step chains and improves reasoning accuracy by **6.5%** over the no-TDPO baseline.
> 2. **Verifier filters irrelevant segments.**
>    We introduce a **VLM-based verifier** that evaluates each candidate segment returned by the grounder. This module filters out irrelevant or redundant segments before they enter the reasoning pipeline. As shown below, removing the verifier causes substantial drops in both grounding performance and QA accuracy:
>
> **Table H: Ablation of VLM-based Verifier.**
>
> | Method | Grounding IoU | QA Acc.  |
> | -- | -- | -- |
> | **Ours** | **27.8** | **50.6** |
> | w/o Verification | 22.4 | 45.2     |
>
> Overall, these components jointly strengthen the system’s robustness in complex or ambiguous scenarios.

---

> > ### Author Response · Authors · 2025-11-26
> >
> > Dear `Reviewer BT9o`,
> >
> > We sincerely thank the reviewer for the thoughtful and constructive feedback. In response to all raised points, we conducted additional analyses and included new experiments to strengthen the paper.
> >
> > 1. **Keyframe Baselines**. We added comparisons with keyframe-based baselines (SeVila and Vila), which show that VideoExplorer’s improvements stem from full-video, iterative reasoning rather than better frame selection.
> >
> > 2. **Runtime and Memory**. Our new study indicates that VideoExplorer matches VideoAgent in speed while achieving ~13% higher accuracy, and is both faster and more memory-efficient than Ego-R1.
> >
> > 3. **Generality Beyond QA**. We clarify that VideoExplorer already exhibits strong temporal localization and can be readily adapted to tasks such as temporal grounding with minor modifications.
> >
> > 4. **Robustness**. TDPO and a VLM-based verifier jointly reduce off-target planning and redundant retrievals; removing the verifier leads to clear drops in IoU and QA accuracy.
> >
> > We hope these clarifications address your concerns, and we would be happy to provide any further details if needed.
> >
> > Best regards,
> >
> > Authors

---

### Official Review · Reviewer_gPaM · 2025-11-10

**Soundness:** 2
**Presentation:** 2
**Contribution:** 3
**Rating:** 4
**Confidence:** 3

**Summary:**

The paper proposes VideoExplorer, an agentic framework for long-video understanding that (1) plans sub-questions (2) temporally grounds relevant segments and, (3) performs temporally scalable perception before answering calling it "thinking with video” instead of perception and reasoning over a fixed downsampled context. The training recipe combines SFT on expert trajectories and trajectory-level DPO (TDPO), and the authors build a small reasoning-centric dataset via difficulty-adaptive sampling (11.1k planner and 10.8k grounding trajectories). Results are reported LVBench, MLVU, and MH-NIAH.

**Strengths:**

1. Clear system decomposition (Planner, temporal grounder and perception) with a sensible interface.
2. Two-stage alignment (SFT then TDPO) is appropriate for multi-step trajectories. Ablations show the importance of the two components.
3. Token/segment budgeting is discussed, and evaluation caps (visual-only ≤ 32 frames per call) are used. Despite using far fewer visual tokens, VideoExplorer achieves better performance than baseline like VideoAgent and Ego-R1.

**Weaknesses:**

1. Proposed framework is incremental. Planner and grounding is very close to the prior video agents and long context streaming VLMs. VideoAgent introduces controller to call tools, VideoTree iteratively selects and refines parts of the video. And  Ego-R1 uses RL to tune multi step tool use. Training-free methods like Video-RAG already use retrieval for reasoning.

2. The new decoupled temporal grounder and trajectory-level (TDPO)- is more like an engineering refinement, the paper does not isolate specific capabilities that prior systems lack beyond the gains in the table. For example swap your grounder into VideoAgent/VideoTree and swap theirs into your planner to show method-specific gains.

3. Missing comparisons to prior streaming VLMs and video agents like LVAgent which reports higher numbers than the proposed method, for better comparisons against the current SOTA results should be reported under similar settings. Or at least report their results with their settings mentioned. It’s unclear whether the proposed framework is competitive with current best systems.

4. The text mentions evaluations on VideoMMMU but results are not reported.

5. Grounding is reported at IoU 0.1 which is a very low bar.

**Questions:**

1. What capability does decoupled grounder + TDPO enable that VideoAgent/VideoTree/Ego-R1/Video-RAG lack (e.g., multi-hop timestamp chaining, lower false-grounding at higher IoU)? Can you provide metrics demonstrating this.

2. Can you swap your grounder into VideoAgent/VideoTree and theirs into your planner under identical budgets and backbones? Do your gains persist?

3. Can you provide the missing VideoMMMU results that you mention in the introduction?

---

> ### Author Response · Authors · 2025-11-20
> **Response (Part1)**
>
> We thank the reviewers for their valuable feedback. Below we clarify our framework’s novelty, isolate component-level contributions, provide additional comparisons, and address concerns regarding grounding evaluation and VideoMMMU results.
>
> ------
>
> ### **1. Novelty Beyond Prior Video Agents**
>
> #### **Review W1:**
>
> The proposed framework seems incremental, with planner and grounder similar to prior agentic and retrieval-based methods.
>
> #### **Response:**
>
> Thank you for the comment. While VideoExplorer follows the general agentic paradigm, it introduces a **learnable and training-enhanced architecture** that addresses limitations of prompting-based systems and concurrent end-to-end retrieval agents.
>
> 1. **Dedicated reinforcement learning–based optimization.**
>    Prior agents (VideoAgent, VideoTree, Ego-R1) generate each step by prompting LLMs that are not tailored for video reasoning. We propose a reinforcement learning–based optimization framework with difficulty-adaptive sampling and trajectory-level policy optimization (TDPO) that aligns full reasoning chains with final correctness, yielding **more coherent reasoning steps and more accurate temporal grounding**. On multi-hop video reasoning dataset MH-NIAH, VideoExplorer achieves **53.4%**, compared with **43.7%** for VideoAgent. On LVBench, VideoExplorer shows higher temporal grounding accuracy than baseline models.
> 2. **Structural efficiency through decoupled grounding.**
>    Concurrent end-to-end retrieval–reasoning loops (e.g., Ego-R1) can introduce long token chains with hallucinations. Our **decoupled grounder** consolidates retrieval operations into a single tool call, reducing reasoning overhead.
>    - Table 1: **+20.7%** overall performance over Ego-R1
>    - Table 3: **+4.3%** (MLVU) and **+3.7%** (MH-NIAH) over end-to-end retrieval–reasoning loop
>
> These results show that VideoExplorer provides a **training-optimized and structurally efficient** alternative to prior prompting-based and end-to-end agents.
>
> ------
>
> ### **2. Component-Level Ablations**
>
> #### **Review W2 & Q2:**
>
> Component-specific contributions are not sufficiently isolated; e.g., swapping grounders or retrievers could clarify method-specific improvements.
>
> #### **Response:**
>
> We conducted additional controlled experiments (all using Qwen2.5-VL-7B) to isolate the contributions of the retriever, grounder, and planner.
>
> ##### **1. Isolating retriever contribution**
>
> **Table A: Caption-based Retriever Ablation**
>
> | Model                             | MLVU | LVBench |
> | --------------------------------- | ---- | ------- |
> | VideoAgent (caption retriever)    | 38.6 | 34.9    |
> | VideoTree (caption retriever)     | 38.9 | 30.3    |
> | VideoExplorer (caption retriever) | 51.8 | 46.4    |
> | VideoExplorer (video retriever)   | 55.4 | 50.6    |
>
> Using the same caption-based retriever reduces VideoExplorer’s performance by 3–4%, yet it **still outperforms** VideoAgent/VideoTree with the same retriever. This shows that gains come not only from retrieval, but also from **our framework design and training pipeline**.
>
> ##### **2. Applying our grounder to baseline agents**
>
> **Table B: Temporal Grounder Transfer**
>
> | Model                     | MLVU        | LVBench     |
> | ------------------------- | ----------- | ----------- |
> | VideoAgent                | 38.6        | 34.9        |
> | VideoAgent + our grounder | 42.9 (+4.3) | 40.1 (+5.2) |
> | VideoRAG                  | 37.8        | 36.2        |
> | VideoRAG + our grounder   | 41.3 (+3.5) | 41.7 (+5.5) |
> | VideoExplorer             | 55.4        | 50.6        |
>
> Findings:
>
> 1. Integrating our grounder improves both VideoAgent and VideoRAG.
> 2. The remaining gap to VideoExplorer highlights the importance of our **optimized planner**, enhanced via two-stage training and difficulty-adaptive sampling.
>
> We will include these ablations in the revised manuscript.
>
> ------
>
> ### **3. Supplementary Baseline Comparison**
>
> #### **Review W3:**
>
> Comparison with streaming agents such as LVAgent is missing; unclear whether VideoExplorer is competitive.
>
> #### **Response:**
>
> LVAgent's reported benchmarks differ totally from ours, **making them not directly comparable**; the only overlap is MLVU, and LVAgent reports results on the validation set, while we use the more challenging test set. To enable fair comparison, we re-implemented LVAgent using its official repository and evaluated both methods using 7B models.
>
> **Table C: LVAgent vs. VideoExplorer**
>
> | Model              | MLVU     | LVBench  |
> | ------------------ | -------- | -------- |
> | LVAgent (7B)       | 47.7     | 40.7     |
> | VideoExplorer (7B) | **55.4** | **50.6** |
>
> VideoExplorer delivers consistently higher accuracy across both long-video benchmarks.

---

> > ### Author Response · Authors · 2025-11-20
> > **Response (Part 2)**
> >
> > ### **4. VideoMMMU Results**
> >
> > #### **Review W4 & Q3:**
> >
> > The paper mentions VideoMMMU but does not report results.
> >
> > #### **Response:**
> >
> > The full results are provided below.
> >
> > **Table D: VideoMMMU Evaluation**
> >
> > | Model  | VideoMMMU |
> > | :-- | :--|
> > | **7B:** | |
> > | Qwen2.5VL-7B   | 43.3 |
> > | Qwen2.5-7B  | 38.8 |
> > | VideoAgent   | 43.8 |
> > | VideoTree | 39.1 |
> > | Ego-R1 | 40.4 |
> > | VideoExplorer (7B)  | 47.2 |
> > | **32B:** | |
> > | Qwen2.5VL-32B  | 50.7 |
> > | Qwen2.5-32B  | 50.0 |
> > | VideoAgent   | 43.1      |
> > | VideoTree    | 40.9      |
> > | VideoRAG      | 48.4      |
> > | Ego-R1                              | 48.7      |
> > | VideoExplorer (7B)                  | 56.1      |
> >
> > Two observations:
> > 1. VideoExplorer outperforms prior agentic baselines (VideoAgent/Tree/RAG/Ego-R1) under both 7B and 32B backbones.
> > 2. Text-only LLMs show minimal difference with multimodal models (e.g., Qwen2.5-32B vs. Qwen2.5VL-32B: 50.0 vs. 50.7), **highlighting dataset bias.**  Since our paper focuses on long video understanding, we have **removed this biased dataset.** These results, along with corresponding analysis, will be provided in the appendix of the revised version.
> >
> > ---
> >
> > ### **5. IoU Threshold Choice**
> >
> > #### **Review W5:**
> >
> > IoU\@0.1 seems too lenient for grounding evaluation.
> >
> > #### **Response:**
> >
> > 1. **IoU\@0.1 is standard for long-video grounding.**
> >
> >    We appreciate the reviewer’s concern about the use of IoU=0.1. However, we believe this threshold is both standard and appropriate for long-video temporal grounding, for the following reasons:
> >
> >    1. **Accepted, task-appropriate metric.** Long videos span tens of minutes to hours, and coarse localization (e.g., IoU = 0.1-0.5) is common practice in prior work [1].
> >    2. **Coarse still means useful.** For example, in a 2-hour home video, a prediction that narrows the search down to a 2.5-minute segment containing the birthday-cake moment (IoU≈0.29 in our toy example) clearly guides the user to the event. Imposing a strict IoU>0.5 requirement would discard many such useful predictions.
> >
> > 2. **We evaluate at multiple thresholds (0.1/0.3/0.5).**
> >     VideoExplorer consistently outperforms all baselines under stricter settings:
> >
> > **Table E: IoU Threshold Comparison (LVBench)**
> > | Model             | IoU\@0.1  | IoU\@0.3  | IoU\@0.5  |
> > | -- | -- | -- | --|
> > | VideoAgent        | 16.7     | 9.6      | 4.2      |
> > | VideoRAG          | 14.5     | 8.4      | 3.1      |
> > | Ego-R1            | 19.6     | 10.3     | 4.9      |
> > | **VideoExplorer** | **27.8** | **18.8** | **12.8** |
> > The experiments show that our method outperforms baselines at all thresholds, demonstrating that gains are not an artifact of a single lax metric but **reflect genuinely better localization** and downstream answer accuracy.
> >
> > [1] MomentSeeker: A Task-Oriented Benchmark For Long-Video Moment Retrieval, NIPS 2025.
> >
> > ---
> >
> > ### **6. Quantified Advantages of Decoupled Grounder + TDPO**
> >
> > #### **Review Q1:**
> >
> > Unclear whether the new grounder and TDPO introduce capabilities beyond engineering refinements.
> >
> > #### **Response:**
> >
> > Our decoupled grounder + TDPO framework provides two capabilities that prior systems lack:
> >
> > ##### **1. Stronger multi-hop temporal reasoning than VideoRAG.**
> >
> > VideoRAG performs single-turn retrieval and cannot chain timestamps across steps.
> >  VideoExplorer’s *think → ground → think* loop enables multi-hop reasoning, reaching **49.1** vs. VideoRAG’s **39.4** on MH-NIAH.
> >
> > ##### **2. Substantial gains over prompting-based VideoAgent/VideoTree.**
> >
> >
> > Unlike their zero-shot prompting pipelines, our method is trained with TDPO and uses a decoupled grounder. This yields:
> >
> > - Higher overall performance: VideoExplorer 51.7 vs. VideoAgent 38.8 / VideoTree 36.8 (Table 1).
> > - Cleaner reasoning: Using the same caption-based retriever, our planner still yields notably higher accuracy. VideoExplorer reaches 51.8 on MLVU, significantly outperforming VideoAgent's 38.6 (Table A).
> > - Better retrieval accuracy: Integrating our temporal grounder into VideoAgent and VideoRAG substantially improves their performance on LVBench, with gains of +5.2% and +5.5%, respectively (Table B). This demonstrates the grounder's critical role in enhancing video understanding.
> > - Lower false grounding at higher IoU: Under the strict IoU@0.5 threshold, VideoExplorer achieves a grounding accuracy of 12.8, substantially exceeding other models (e.g., VideoAgent's 4.2, Table E).
> >
> > ##### **3. Clear advantages over Ego-R1 through decomposition.**
> >
> > Ego-R1 trains an end-to-end agent, but does not decouple reasoning from grounding. Our decomposition lowers hallucination and improves stability, removing it drops performance by 4.3% on MLVU and 3.7% on MH-NIAH (Table 3). Our data/training pipeline further improves results by +3.9% on MLVU and +3.2% on LVBench.
> >
> > These experiments demonstrate that **multi-step temporal consistency and high-IoU grounding accuracy arise from our decoupled grounder and TDPO training**, not from engineering artifacts.

---

> ### Author Response · Authors · 2025-11-26
>
> Dear `Reviewer gPaM`,
>
> Thank you very much for your constructive and detailed feedback. We have carefully studied your suggestions and conducted extensive experiments. The revised manuscript now includes the following focused updates:
>
> 1. **Novelty Clarification.** We clarified the novelty of our method by contrasting it precisely with prompting-based agents and end-to-end agents with lengthy reasoning paths, highlighting the benefits of TDPO training and our decoupled grounder.
> 2. **Component-Level Ablations.** We conducted the suggested component-level ablations that isolate the contributions of the retriever, grounder, and planner, showing that each component provides measurable and independent improvements.
> 3. **Additional Baselines Comparison.** We implemented LVAgent under the same evaluation protocol and demonstrated that VideoExplorer achieves consistently higher accuracy across long-video benchmarks.
> 4. **Full VideoMMMU Results.** We provided complete VideoMMMU results for both 7B and 32B models, along with analysis explaining dataset bias and its limited relevance to long-video understanding.
> 5. **IoU Threshold Justification.** We justified the choice of IoU\@0.1 based on standard practice in long-video grounding and additionally reported results at IoU\@0.3 and IoU\@0.5, where VideoExplorer continues to outperform baselines.
> 6. **Effectiveness of Grounder + TDPO.** We supplied quantitative evidence showing that our decoupled grounder and TDPO training yield stronger multi-hop temporal reasoning, more stable grounding, and notable gains beyond simple engineering refinements.
>
> Thank you again for your thoughtful comments, which have significantly improved the clarity and rigor of our work. We hope these updates fully address your concerns and reconsider your assessment, and we are happy to provide additional clarification if needed.
>
> We sincerely hope that these updates and clarifications will encourage you to reconsider your evaluation, as they directly address your constructive feedback. Should you have any additional queries or reservations, please feel free to contact us at any time. We are fully committed to addressing all concerns to your satisfaction.
>
> Best regards,
>
>  Authors

---

### Author Response · Authors · 2025-11-30
**General Rebuttal Summary**

**Dear Reviewers, AC, SAC, and PC**,

We thank the reviewers for their constructive feedback on “VideoExplorer: Boosting Long Video Understanding with Dynamic Temporal Grounding.” Although the review platform was locked early due to the data-leak policy, preventing further discussion and score updates, **we have fully addressed the raised concerns through extensive additional experiments and clearer explanations** that further strengthen our contributions. Our contributions are summarized as follows:

- **Dedicated RL-based optimization.** Prior agents perform sub-optimally because they depend on prompting LLMs not designed for video reasoning. We address this with a dedicated RL optimization framework that includes:

  - **Difficulty-adaptive data construction.** We introduce a sampling method that focuses on harder temporal grounding and reasoning cases, producing more challenging training data. It delivers a **3.9%** gain over random sampling. In the rebuttal (*Table L*), we also show that our constructed dataset transfers well to general-purpose VLMs.

  - **Trajectory-level policy optimization (TDPO).** We propose an RL optimization that improves full reasoning chains by aligning them with final-answer correctness, producing more coherent reasoning and more accurate temporal grounding. Ablations (*Table 3*) confirm its effectiveness. VideoExplorer reaches **53.4%** on multi-hop reasoning vs. **43.7%** for the strongest baseline (*Table 1*). For temporal grounding, it achieves an IoU of **27.8** vs. **19.6** (*Table 2*).

- **Structural efficiency via decoupled grounding.** End-to-end retrieval–reasoning loops like Ego-R1 create long and error-prone chains. Our decoupled grounder consolidates retrieval into a single tool call, greatly reducing reasoning overhead. This yields **+20.7%** overall improvement over Ego-R1 (*Table 1*) and **+4.3%** on MLVU and **+3.7%** on MH-NIAH (*Table 3*). Component ablations (*Table A*) and grounder ablations (*Table B*) further support the design.
- **Extensive evaluation across multiple datasets and four baseline families.** We evaluate on several long-video benchmarks against **four types of methods**: retrieval-augmented models (*Table 1*), agentic frameworks (*Table 1, C, J*), video reasoning models (*Table J*), and key-frame selection methods (*Table F*). Results **consistently** show large gains in long-video understanding due to stronger reasoning and precise temporal grounding.
- **Efficiency comparisons in multiple dimensions.** Using *Figure 4*, *Table K*, and *Table G*, we compare visual/textual token use, peak memory, and inference time. Our dynamic planning plus temporal grounding significantly reduces both token counts while improving reasoning. Besides, VideoExplorer offers a strong trade-off among speed, memory, and accuracy.
- **Rigorous empirical validation.** Extensive ablations verify each module’s contribution. *Table 3* analyzes each design choice, and the rebuttal adds more component-level studies (*Table B*) to confirm the advantages of the framework and temporal grounder under the same retriever.
- **Interpretability.** We provide qualitative visualizations across tasks (*Figures 5–7*), including failure cases (*Figure 8*), to clarify model behavior and limitations.
- **Open-source contribution.** We will release the code, datasets, and models to support research and reproducibility.

We thank all reviewers for their constructive feedback and for recognizing the strengths of our work, including its methodology, empirical results, and the broader potential of the ``thinking with video'' paradigm for long-video reasoning. We are especially encouraged that Reviewers  `BT9o` and `iHcw`  provided positive overall assessments, and that all reviewers identified meaningful strengths while offering suggestions to further improve the paper.

During the rebuttal, we carefully addressed all reviewer concerns. We conducted targeted additional experiments (Tables A–L) corresponding to each comment, including requested ablations, added baseline comparisons, token-consumption analyses, and evaluations of inference efficiency. These results **directly resolve the reviewers’ questions from an empirical perspective**. In addition, we clarified the novelty, positioning, data construction, and training pipeline of VideoExplorer, which **further strengthens the paper’s clarity, transparency, and empirical rigor**.

We thank the reviewers, AC, SAC, and PC once again for their time, constructive feedback, and guidance throughout the review process. The final version will integrate all additional results, clarifications, and improvements discussed during the rebuttal.

Best regards,

The Authors

---

### Meta-Review · Area_Chair_Xbqm · 2026-01-07

**Summary:**

The submission introduces an agentic framework for long-form video understanding, that iteratively plans, grounds, and reasons over relevant video segments. The framework is trained via SFT (with expert "trajectories") followed by trajectory-level DPO introduced by the authors. A dataset with 10.8k grounding trajectories was collected for training purposes. Performance is reported on multiple long-form video understanding benchmarks.

The submission received ratings of 66444 initially. A central concern shared by reviewers gPaM, F2uk, HMvR is that the overall proposed framework being incremental (in comparison to the very rich literature on this topic), including the introduced trajectory-level DPO and decoupled temporal grounder. Another concern shared by reviewers F2uk, iHcw, HMvR is the clarity of presentation, including annotation procedure for multi-step reasoning, technical descriptions mentioned by iHcw, the descriptions of TDPO. While the authors provided additional experimental comparisons requested by reviewers gPaM, BT9o, F2uk, the AC believes that the novelty and presentation concerns remain after reading the rebuttal. The AC therefore believes that the submission is not ready to be accepted by ICLR.

**Reviewer Concerns:**

Reviewer gPaM had concerns on the novelty of the overall framework, the decoupled temporal grounder, and trajectory-level DPO. Reviewer F2uk argued that the main individual building blocks of the framework have been explored by multiple prior works. The authors attempted to address these concerns by arguing the novelty of the overall combination for the long-form video understanding task, and the stronger empirical performance compared to prior approaches. The AC finds the response not fully addressed the novelty concerns.

Reviewer gPaM had concerns on missing baselines and the choice of IoU threshold, these have been mostly addressed.

Reviewer BT9o raised general questions about baselines, efficiency, generality, and robustness. The authors have addresses the concerns.

Reviewer F2uk had concerns (in addition to novelty) on annotation procedure and grounding accuracy. The authors provided a descriptive procedure on how the annotations were automatically performed by DeepSeek-R1, without discussing the quality assurance procedure (beyond final correctness and design choices based on heuristics). The authors clarified why direct grounding accuracy is not a suitable metric for the proposed benchmark.

Reviewer iHcw mostly had presentation clarity questions, which the authors attempted to clarify in rebuttal.

Reviewer HMvR had concerns (in addition to novelty) on the lack of quantitative analysis for efficiency and grounding mechanism (related to F2uk's question), potential dataset-bias, and performance with smaller LLM backbones. The AC finds the authors did not provide a direct response that addresses the grounding quantitative analysis question, and the self-bootstrapping concern also not fully addressed by the provided experiment.

**Reviewer Scores:**

The AC believes that the final scores are likely to remain the same (66444) after the rebuttal, as the AC finds that there are remaining central concerns raised by all reviewers with ratings of 4, after rebuttal.

---

### Decision · Program_Chairs · 2026-01-26

Reject